# VALUE FLOWS

**Perry Dong**[*1]     **Chongyi Zheng**[*2]

**Chelsea Finn**[1]     **Dorsa Sadigh**[1]     **Benjamin Eysenbach**[2]

[1]Stanford University     [2]Princeton University

perryd@stanford.edu     chongyiz@princeton.edu

## ABSTRACT

While most reinforcement learning methods today flatten the distribution of future returns to a single scalar value, distributional RL methods exploit the return distribution to provide stronger learning signals and to enable applications in exploration and safe RL. While the predominant method for estimating the return distribution is by modeling it as a categorical distribution over discrete bins or estimating a finite number of quantiles, such approaches leave unanswered questions about the fine-grained structure of the return distribution and about how to distinguish states with high return uncertainty for decision-making. The key idea in this paper is to use modern, flexible flow-based models to estimate the full future return distributions and identify those states with high return variance. We do so by formulating a new flow-matching objective that generates probability density paths satisfying the distributional Bellman equation. Building upon the learned flow models, we estimate the return uncertainty of distinct states using a new flow derivative ODE. We additionally use this uncertainty information to prioritize learning a more accurate return estimation on certain transitions. We compare our method (Value Flows) with prior methods in the offline and online-to-online settings. Experiments on 37 state-based and 25 image-based benchmark tasks demonstrate that Value Flows achieves a $1.3\times$ improvement on average in success rates.

Website: https://pd-perry.github.io/value-flows

Code: https://github.com/chongyi-zheng/value-flows

## 1 INTRODUCTION

While many of the recent successes in reinforcement learning (RL) have focused on estimating future returns as a single scalar, modeling the entire future return distribution can provide stronger learning signals and indicate bits about uncertainty in decision-making. Distributional RL promises to capture statistics of future returns, achieving both strong convergence guarantees (Bellemare et al., 2017; Wang et al., 2023a; 2024) and good performance on benchmarks such as Atari and D4RL (Bellemare et al., 2017; Dabney et al., 2018; Ma et al., 2021). This paper aims to understand the benefits of using a more flexible representation of the return distribution, both as a critic in actor-critic RL and for estimating the variance in the future returns.

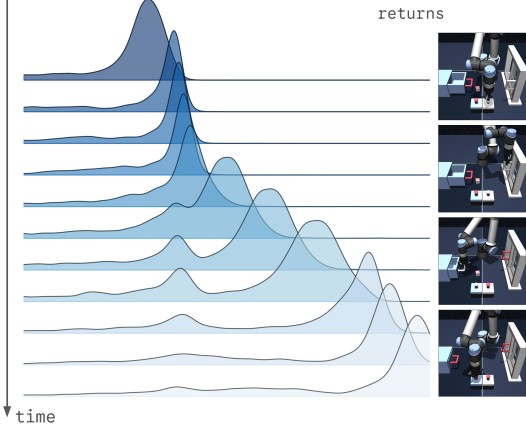

Figure 1: **Value Flows** models the return distribution at each time step using a flow-matching model that is optimized to obey the Bellman equation at each transition.

Modern generative models, such as flow matching (Lipman et al., 2023; 2024; Liu et al., 2023; Albergo & Vanden-Eijnden, 2023) and diffusion models (Sohl-Dickstein et al., 2015; Ho et al., 2020; Song et al., 2021), have shown great success in representing complex, multi-modal distributions in continuous spaces. Building upon their successes,

---

*Equal contribution. Author order was determined by a coin flip.

we will use expressive flow-based models to fit the return distributions and estimate the return uncertainty of different states. These bits of uncertainty information, in turn, allow us to prioritize learning the return estimation on certain transitions.

The main contribution of our work is Value Flows, a framework for modeling the return distribution using flow-matching methods (Fig. 1). The key idea is to formulate a distributional flow-matching objective that generates probability density paths satisfying the distributional Bellman equation automatically. Using the learned flow-based models, we estimate the return variance of distinct states using a new flow derivative ODE and use it to reweight the flow-matching objective. Value Flows also enables efficient estimation of the return expectation (Q-value), bridging a variety of policy extraction methods. Experiments on 37 state-based and 25 image-based benchmark tasks show Value Flows outperforms alternative offline and offline-to-online methods by $1.3\times$ improvement on average.

## 2  RELATED WORK

Distributional RL views the RL problem through the lens of the future return distribution instead of a scalar Q-value (Engel et al., 2005; Muller et al., 2024; Morimura et al., 2010). Prior work has shown both strong convergence guarantees (Wang et al., 2023a; 2024) and good empirical performance (Farebrother et al., 2024; Bellemare et al., 2017; Ma et al., 2025) of this family of methods, enabling applications in exploration (Mavrin et al., 2019) and safe RL (Ma et al., 2021; 2025). However, those methods typically parameterize the return distribution as discrete bins and minimize the KL divergence to the distributional Bellman target (Bellemare et al., 2017) or use a finite number of quantiles to approximate the full return distribution (Dabney et al., 2018). In contrast, Value Flows models the full return distribution directly using state-of-the-art generative models.

Perhaps the most popular modern generative models are autoencoders (Kingma & Welling, 2013), autoregressive models (Vaswani et al., 2017), denoising diffusion models (Sohl-Dickstein et al., 2015; Ho et al., 2020; Song et al., 2021), and flow-matching models (Lipman et al., 2023; 2024; Liu et al., 2023; Albergo & Vanden-Eijnden, 2023). In the context of RL, these generative models have succeeded in modeling the trajectories (Chen et al., 2021; Ajay et al., 2023), the transitions (Alonso et al., 2024; Janner et al., 2021; Farebrother et al., 2025b), the skills (Ajay et al., 2021; Pertsch et al., 2021; Frans et al., 2024; Zheng et al., 2025), and the policies (Wang et al., 2023b; Hansen-Estruch et al., 2023b; Dong et al., 2025c; Park et al., 2025c; Dong et al., 2025b). We employ the state-of-the-art flow-matching objective (Lipman et al., 2023) to estimate the future return distribution.

This paper solves offline RL and offline-to-online RL problems. The goal of offline RL is to learn a policy from previously collected datasets, often through conservative behavioral regularization (Peng et al., 2019; Fujimoto & Gu, 2021; Kostrikov et al., 2021a; Hansen-Estruch et al., 2023b; Park et al., 2025c) or value constraints (Kumar et al., 2020b; Kostrikov et al., 2021b). After learning the offline policy, offline-to-online RL uses interactions with the environment to fine-tune an online policy, often requiring balancing exploration (Yang et al., 2023; Mark et al., 2023), calibrating values (Nakamoto et al., 2024), or maintaining offline data (Nair et al., 2021; Ball et al., 2023; Dong et al., 2025a;b). Our approach focuses on estimating the full return distribution, resulting in more accurate Q-value estimations that benefit both settings.

Prior work has used confidence weight from Q estimates to reweight the Bellman error (Kumar et al., 2020a; 2019; Lee et al., 2021; Schaul et al., 2015), aiming to mitigate the issue of instability in Q-learning (Tsitsiklis & Roy, 1997; Baird, 1995; van Hasselt et al., 2015). These methods construct the uncertainty weight using bootstrapped errors from Q estimations (Kumar et al., 2020a; Schaul et al., 2015) or the epistemic uncertainty from an ensemble of Q networks (Lee et al., 2021). Our work builds on these prior works by using variance to reweight the critic objective (a flow-matching objective in our case), and is most similar to prior methods that do weighting based on aleatoric uncertainty (Kumar et al., 2020a; Schaul et al., 2015).

## 3  PRELIMINARIES

We consider a Markov decision process (MDP) (Sutton et al., 1998; Puterman, 2014) defined by a state space $\mathcal{S}$, an action space $\mathcal{A} \subset \mathbb{R}^d$, an initial state distribution $\rho \in \Delta(\mathcal{S})$, a *bounded* reward

function $r : \mathcal{S} \times \mathcal{A} \rightarrow [r_{\min}, r_{\max}]$,[1] a discount factor $\gamma \in [0, 1)$, and a transition distribution $p : \mathcal{S} \times \mathcal{A} \rightarrow \Delta(\mathcal{S})$, where $\Delta(\mathcal{X})$ denotes the set of all possible probability distributions over a space $\mathcal{X}$. We will use $h$ to denote the time step in an MDP, use uppercase $X$ to denote a random variable, use lowercase $x$ to denote the value of $X$, use $F_X$ to denote the cumulative distribution function (CDF) of $X$, and use $p_X$ to denote the probability density function (PDF) of $X$. In Appendix A.1, we include a brief discussion about the expected discounted return and the Q function in RL.

This work considers the offline RL problems (Lange et al., 2012) (Sec. 4.4), which aim to maximize the return using a fixed dataset. We will use $D = \{(s, a, r, s')\}$ to denote the dataset with transitions collected by some behavioral policies. We also extend our discussions to offline-to-online RL problems (Sec. 4.4), where algorithms first pre-train a policy from the offline dataset $D$ and then fine-tune the policy with online interactions in the exact *same* environment. Those online interactions will also be stored in $D$, enabling off-policy learning.

**Distributional RL.** Instead of focusing on the expected discounted return (scalar), distributional RL (Morimura et al., 2010; Bellemare et al., 2017) models the entire return distribution. Given a policy $\pi$, we denote the (discounted) return random variable as $Z^\pi = \sum_{h=0}^\infty \gamma^h r(S_h, A_h) \in \left[ z_{\max} \triangleq \frac{r_{\min}}{1-\gamma}, z_{\min} \triangleq \frac{r_{\max}}{1-\gamma} \right]$, and denote the conditional return random variable as $Z^\pi(s, a) = r(s, a) + \sum_{h=1}^\infty \gamma^h r(S_h, A_h)$.[2] We note that the expectation of the conditional return random variable $Z^\pi(s, a)$ is equivalent to the Q function: $Q^\pi(s, a) = \mathbb{E}_{\pi(S_0=s, A_0=a, S_1, A_1, \cdots)} [Z^\pi(s, a)]$.

Prior work (Bellemare et al., 2017) defines the distributional Bellman operator under policy $\pi$ for a conditional random variable $Z(s, a)$ as

$$\mathcal{T}^\pi Z(s, a) \stackrel{d}{=} r(s, a) + \gamma Z(S', A'), \tag{1}$$

where $S'$ and $A'$ are random variables following the joint density $p(s' \mid s, a)\pi(a' \mid s')$ and $\stackrel{d}{=}$ denotes identity in distribution. This definition implies two important relationships for the CDF and the PDF of the random variable $\mathcal{T}^\pi Z(s, a)$ (see Lemma 1 and Lemma 2 in Appendix A.2). We will use these two implications to derive our method in Sec. 4.1 and Sec. 4.2. Prior work (Bellemare et al., 2017) has already shown that the distributional Bellman operator $\mathcal{T}^\pi$ is a $\gamma$-contraction under the $p$-Wasserstein distance, indicating that $\mathcal{T}^\pi$ has a unique fixed point. We include a justification for the fixed point $Z^\pi(s, a)$ in Lemma 3, and will use the contraction property of $\mathcal{T}^\pi$ to estimate the return distribution with theoretical guarantees (Sec. 4.1 & 4.2).

**Flow matching.** Flow matching (Lipman et al., 2023; 2024; Liu et al., 2023; Albergo & Vanden-Eijnden, 2023) refers to a family of generative models based on ordinary differential equations (ODEs). The goal of flow matching methods is to transform a simple noise distribution into a target distribution $p_\mathcal{X}$ over some space $\mathcal{X} \subset \mathbb{R}^d$. We will use $t$ to denote the flow time step and sample the noise $\epsilon$ from a standard Gaussian distribution $\mathcal{N}(0, I)$ throughout our discussions. Flow matching uses a time-dependent vector field $v : [0, 1] \times \mathbb{R}^d \rightarrow \mathbb{R}^d$ to construct a time-dependent diffeomorphic flow $\phi : [0, 1] \times \mathbb{R}^d \rightarrow \mathbb{R}^d$ (Lipman et al., 2023; 2024) that realizes the transformation. The resulting probability density path $p : [0, 1] \times \mathbb{R}^d \rightarrow \Delta(\mathbb{R}^d)$ of the vector field $v$ satisfies the continuity equation (Lipman et al., 2023). Appendix A.3 includes the flow ODE (Eq. 11) and the continuity equation (Sec. 12) connecting $\phi$, $v$ and $p$. In Sec. 4, we will estimate the return distribution using flow-based models, utilizing the continuity equation (Sec. 4.2) and flow ODE (Sec. 4.3).

Prior work has proposed various formulations for learning the vector field (Lipman et al., 2023; Liu et al., 2023; Albergo & Vanden-Eijnden, 2023). We adopt the simplest flow matching objectives called conditional flow matching (CFM) (Lipman et al., 2023) to construct loss functions for optimizing the return vector field (Sec. 4.2). Appendix A.3 includes detailed discussions of flow matching methods.

## 4 VALUE FLOWS

In this section, we introduce our method for efficiently estimating the return distribution using an expressive flow-matching model, resulting in an algorithm called Value Flows. We first formalize the

---

[1] While we consider deterministic rewards, our discussions generalize to stochastic rewards used in prior work (Bellemare et al., 2017; Dabney et al., 2018; Ma et al., 2021).

[2] By definition, we can also write $Z^\pi = Z^\pi(S_0, A_0)$.

problem setting, providing motivations and desiderata for the flow-based return estimation. We then introduce the update rules for the return vector field, deriving a distributional flow matching loss to fit the discounted return distribution. Our practical algorithm will build upon this distributional flow matching loss and incorporate uncertainty in the return distribution.

## 4.1 MOTIVATIONS AND DESIDERATA

While prior actor-critic methods (Fujimoto et al., 2018; Haarnoja et al., 2018) typically involve estimating the *scalar* Q function of a policy, we instead consider estimating the entire return distribution using state-of-the-art generative models. Our desiderata are threefold: we want our model capable of *(1)* estimating the full distribution over returns without coarse-graining *(2)* learning probability densities satisfying the distributional Bellman backup (Eq. 9), while preserving convergence guarantees, and *(3)* prioritizing learning a more accurate return estimation at transitions with high variance. Prior distributional RL methods either discretize the return distribution (Bellemare et al., 2017) or use a finite number of quantiles to represent the return distribution (Dabney et al., 2018), violating these desiderata.

To achieve these goals, we use an expressive flow-based model to represent the conditional return random variable $Z^\pi(s, a)$ (desiderata *(1)*). We will learn a time-dependent vector field $v : \mathbb{R} \times [0, 1] \times \mathcal{S} \times \mathcal{A} \to \mathbb{R}$ to model the return distribution. The desired vector field $v^\star$ will produce a time-dependent probability density path $p^\star : \mathbb{R} \times [0, 1] \times \mathcal{S} \times \mathcal{A} \to \Delta(\mathbb{R})$ that satisfies the distributional Bellman equation (Eq. 10) for any flow time step $t \in [0, 1]$,

$$p^\star(z^t \mid t, s, a) = \mathcal{T}^\pi p^\star(z \mid t, s, a) = \frac{1}{\gamma} \mathbb{E}_{p(s' \mid s, a), \pi(a' \mid s')} \left[ p^\star \left( \left. \frac{z^t - r(s, a)}{\gamma} \right| t, s', a' \right) \right]. \quad (2)$$

Thus, $p^\star(z^1 \mid 1, s, a)$ converges to the discounted return distribution $p_{Z^\pi}(z \mid s, a)$ (desiderata *(2)*). To optimize the vector field $v$ toward $v^\star$, Sec. 4.2 will construct a flow matching loss that resembles temporal difference learning (Sutton, 1988). We will show that both the expected return (Q function) and variance of the return are easy to compute using the initial vector field $v(\epsilon \mid 0, s, a)$ and the derivative of the vector field $\partial v / \partial z \in \mathbb{R}$ respectively (desiderata *(3)*; Sec. 4.3). Our practical algorithm (Sec. 4.4) will weight our flow matching loss using the variance estimate.

## 4.2 ESTIMATING THE RETURN DISTRIBUTION USING FLOW-MATCHING

We start by constructing the update rules for the vector field and the probability density path, and then derive the preliminary flow matching losses to fit the return distribution. We will use these preliminary losses to construct the practical loss used in our algorithm (Sec. 4.4).

**The vector field update rule.** Similar to the standard Bellman operator (Agarwal et al., 2019; Puterman, 2014), the distributional Bellman operator preserves convergence guarantees *regardless* of initialization (Bellemare et al., 2017). This property allows us to first construct update rules for a vector field $v(z^t \mid t, s, a)$ and a probability density path $p(z^t \mid t, s, a)$ *separately* and then relate them via the continuity equation (Eq. 12).

Specifically, given a policy $\pi$ and the transition $p(s' \mid s, a)$, we start from a randomly initialized vector field $v_k(z^t \mid t, s, a)$ (iteration $k = 0$) that generates the probability density path $p_k(z^t \mid t, s, a)$. We construct a new vector field and a new probability density path using $v_k$ and $p_k$:

$$p_{k+1}(z^t \mid t, s, a) \triangleq \mathcal{T}^\pi p_k(z^t \mid t, s, a) = \frac{1}{\gamma} \mathbb{E}_{p(s' \mid s, a), \pi(a' \mid s')} \left[ p_k \left( \left. \frac{z^t - r(s, a)}{\gamma} \right| t, s', a' \right) \right],$$

$$v_{k+1}(z^t \mid t, s, a) \triangleq \frac{\frac{1}{\gamma} \mathbb{E}_{p(s' \mid s, a), \pi(a' \mid s')} \left[ p_k \left( \left. \frac{z^t - r(s, a)}{\gamma} \right| t, s', a' \right) v_k \left( \left. \frac{z^t - r(s, a)}{\gamma} \right| t, s', a' \right) \right]}{p_{k+1}(z^t \mid t, s, a)}. \quad (3)$$

Importantly, these definitions only instantiate the functional form of the new vector field $v_{k+1}$ and the new probability density path $p_{k+1}$, missing a connection between them. Establishing an explicit relationship between the new vector field $v_{k+1}$ and the new probability density path $p_{k+1}$ requires using the continuity equation (Eq. 12) between the vector field $v_k$ and its probability density path $p_k$, which we show in the following proposition.

**Proposition 1** (Informal). *Given the vector field $v_k$ that generates the probability density path $p_k$, the new vector field $v_{k+1}$ generates the new probability density path $p_{k+1}$.*

See Appendix B.1 for the detailed statement and a proof. There are two implications of this proposition. First, since the new vector fireld $v_{k+1}$ generates the new probability density path $p_{k+1}$, applying the distributional Bellman operator to the probability density path $p_k$ is equivalent to learning a vector field $v_{k+1}$ that satisfies Eq. 3. This implication allows us to convert the problem of estimating the full return distribution into the problem of learning a vector field, which naturally fits into the flow-matching framework. Second, since the distributional Bellman operator $\mathcal{T}^\pi$ is a $\gamma$-contraction, repeatedly applying $\mathcal{T}^\pi$ to the probability density path $p_k$ guarantees convergence to $p^\star(z^t \mid t, s, a)$ (Eq. 2). Therefore, we can construct a flow matching loss to learn the vector field $v_{k+1}$ with a guarantee to converge toward $v^\star$.

**The distributional flow matching losses.** We now derive the preliminary flow matching losses for estimating the return distribution only using transition samples from the dataset $D$. Given the vector field $v_k$, one simple approach to learn a new vector field $v$ satisfying the update rule in Eq. 3 is to minimize the mean squared error (MSE) between $v$ and $v_{k+1}$. We call this loss the *distributional flow matching* (DFM) loss:

$$\mathcal{L}_{\text{DFM}}(v, v_k) = \mathbb{E}_{\substack{(s,a,r)\sim D, t\sim\text{UNIF}([0,1]) \\ p_{k+1}(z^t|t,s,a)}} \left[ \left( v(z^t \mid t, s, a) - v_{k+1}(z^t \mid t, s, a) \right)^2 \right]. \tag{4}$$

It is easy to verify that $\arg\min_v \mathcal{L}_{\text{DFM}}(v, v_k) = v_{k+1}$ (see Lemma 4 in Appendix B). Similar to the standard flow matching loss (Lipman et al., 2023), this loss function is not practical because of *(1)* the unknown transition probability density $p(s' \mid s, a)$, and *(2)* the intractable integral (the expectation) in the vector field $v_{k+1}$. To tackle the issue of the intractable vector field $v_{k+1}$, we optimize the alternative *distributional conditional flow matching* (DCFM) loss:

$$\mathcal{L}_{\text{DCFM}}(v, v_k) = \mathbb{E}_{\substack{(s,a,r,s')\sim D, t\sim\text{UNIF}([0,1]) \\ a'\sim\pi(a'|s'), z^t\sim\frac{1}{\gamma}p_k\left(\frac{z^t-r}{\gamma}\Big|t,s',a'\right)}} \left[ \left( v(z^t \mid t, s, a) - v_k\left( \frac{z^t - r}{\gamma} \Big| t, s', a' \right) \right)^2 \right]. \tag{5}$$

We can interpret the transformed returns $(z^t - r)/\gamma$ as convolving the probability density path $p_k$ and use a change of variable to simplify this DCFM loss (see Lemma 5 in Appendix B). Unlike the DFM loss $\mathcal{L}_{\text{DFM}}$, the DCFM loss $\mathcal{L}_{\text{DCFM}}$ can be easily estimated via samples from the dataset $D$. Like the connection between the flow matching loss and the conditional flow matching loss in Lipman et al. (2023), the DCFM loss has the same gradient as the DFM loss, indicating that they have the same minimizer.

**Proposition 2** (Informal). *The gradient $\nabla_v \mathcal{L}_{DFM}(v, v_k)$ is the same as the gradient $\nabla_v \mathcal{L}_{CDFM}(v, v_k)$.*

See Appendix B.1 for the detailed statement and a proof. Importantly, optimizing $\mathcal{L}_{\text{DCFM}}$ does not require access to the full transition distribution and prevents taking the intractable integral. $\mathcal{L}_{\text{DCFM}}$ is a TD loss because it corresponds to applying the distributional Bellman operator $\mathcal{T}^\pi$ to the probability density path $p_k$. Furthermore, our flow-based model produces estimations of the return expectation (Q function; Sec. 4.4) and the return variance (aleatoric uncertainty; Sec. 4.3) easily, both of which will be incorporated into our algorithm.

We note that simply optimizing $\mathcal{L}_{\text{DCFM}}(v, v_k)$ might produce a degenerated vector field, e.g., $v = 0$. In particular, our initial experiments suggest that naively optimizing $\mathcal{L}_{\text{DCFM}}(v)$ produced a trivial performance on some tasks. To stabilize learning, we use the return predictions at the next state-action pair $(s', a')$ and the flow time $t = 1$ to construct a bootstrapped target return similar to the TD target in Q-learning (Watkins & Dayan, 1992). The resulting loss function, called *bootstrapped conditional flow matching* (BCFM) loss $\mathcal{L}_{BCFM}$, resembles the standard fitted Q-learning (Riedmiller, 2005). In addition, we use a target vector field $\bar{v}$ to replace the historical vector field $v_k$. Prior work has shown that using a target network prevents the model from collapsing (Grill et al., 2020; Tian et al., 2021). See Appendix C.1 for further discussions.

### 4.3 HARNESSING UNCERTAINTY IN THE RETURN ESTIMATION

One benefit of using the flow-based models to estimate the full return distribution is that we can easily compute the return variance at every state-action pair $(s, a)$. These return variances measure the

aleatoric uncertainty in the MDP, indicating the noise that stems from the environmental transitions. We incorporate the return uncertainty information into our method and use it to prioritize learning a more accurate return distribution at transitions with higher return variance. We first present an estimation of return expectation using the initial vector field and an approximation of return variance using the Taylor expansion. We then introduce a new ODE that relates the derivative of the learned diffeomorphic flow, allowing us to estimate the return variance in practice. Using this variance estimation, we define the confidence weight for reweighting the distributional flow matching losses.

Prior work (Frans et al., 2025) has shown that the learned vector field $v$ (from Sec. 4.2) at the flow time $t = 0$ points toward the dataset mean. We adopt the same idea to estimate the return expectation ($\mathbb{E}[Z^\pi(s,a)]$, i.e., the Q value) using Gaussian noise $\epsilon \sim \mathcal{N}(0,1)$. Additionally, we estimate the return variance $\mathrm{Var}(Z^\pi(s,a))$ using a first-order Taylor approximation on the corresponding (diffeomorphic) flow $\phi : \mathbb{R} \times [0,1] \times \mathcal{S} \times \mathcal{A} \to \mathbb{R}$ of the vector field $v$. Specifically, the flow $\phi(z \mid t, s, a)$ transforms another noise $\epsilon_0 \sim \mathcal{N}(0,1)$ into a return prediction $\phi(\epsilon_0 \mid 1, s, a)$. We linearize the return prediction around noise $\epsilon$ using a Taylor expansion and then compute the variance of this linearization.

**Proposition 3** (Informal). *The initial vector field $v(\epsilon \mid 0, s, a)$ produces an estimate for the return expectation, while the first-order Taylor approximation of the flow $\phi(\epsilon_0 \mid 1, s, a)$ around $\epsilon$ produces an estimate for the return variance,*

$$\widehat{\mathbb{E}}\left[Z^\pi(s,a)\right] = \mathbb{E}_{\epsilon \sim \mathcal{N}(0,1)}[v(\epsilon \mid 0, s, a)], \quad \widehat{\mathrm{Var}}(Z^\pi(s,a)) = \mathbb{E}_{\epsilon \sim \mathcal{N}(0,1)}\left[\left(\frac{\partial \phi}{\partial \epsilon}(\epsilon \mid 1, s, a)\right)^2\right]. \quad (6)$$

See Appendix B.2 for the detailed statement and a proof. In practice, computing the flow derivative $\partial \phi / \partial \epsilon$ requires backpropagating the gradients through the ODE solver, which is unstable and computationally expensive (Park et al., 2025c). One key observation is that the flow derivative $\partial \phi / \partial \epsilon$ follows another *flow derivative ODE*, drawing a connection with the vector field derivative $\partial v / \partial z$. We will use this flow derivative ODE to compute the flow derivative $\partial \phi / \partial \epsilon$ efficiently.

**Proposition 4** (Informal). *The flow derivative $\partial \phi / \partial \epsilon$ and the vector field derivative $\partial v / \partial z$ satisfy a flow derivative ODE.*

See Appendix B.2 for the detailed statement and a proof. This flow derivative ODE, together with the flow ODE (Eq. 11), enables computing both the return prediction and the return variance estimation (Eq. 6) using a numerical solver (Alg. 2) and an efficient vector-Jacobian product (VJP) implementation (JAX Developers, 2025).

We will use the estimated variance to reweight our flow matching losses (Appendix C.1). Since the distributional RL algorithms typically model the aleatoric uncertainty, a high return variance indicates high stochasticity in the environment, requiring fine-grained predictions. Therefore, the goal of our confidence weight is to prioritize optimizing the vector field at state-action pairs with high return uncertainty. Adapting the confidence weight in prior work (Lee et al., 2021), we define the confidence weight for a state-action pair $(s, a)$ and a noise $\epsilon$ as

$$w(s, a, \epsilon) = \sigma\left(-\tau \left/ \left|\frac{\partial \phi}{\partial \epsilon}(\epsilon \mid 1, s, a)\right|\right.\right) + 0.5, \quad (7)$$

where $\sigma(\cdot)$ denotes the sigmoid function, $\tau > 0$ is a temperature, and $w(s, a, \epsilon) \in [0.5, 1]$. For simplicity, we can use *one* Gaussian noise $\epsilon$ as a Monte Carlo estimator for both the return expectation and the return variance (Eq. 6). We include a visualization of the confidence weight for different temperatures in Appendix D.2. Our confidence weight is an increasing function with respect to the return variance estimate, indicating that a higher uncertainty results in a higher weight. In Appendix C.1, we discuss the complete flow matching losses for fitting the return distribution.

## 4.4 THE COMPLETE ALGORITHM

We now discuss the policy extraction strategies based on the flow-based return models and summarize the complete algorithm of Value Flows. We consider two different behavioral-regularized policy extraction strategies for offline RL and offline-to-online RL. First, for offline RL, following prior work (Li et al., 2025; Chen et al., 2022), we use rejection sampling to maximize Q estimates (Eq. 6) while implicitly imposing a KL constraint (Hilton, 2023) toward a fixed behavioral cloning (BC) policy. Second, for online fine-tuning in offline-to-online RL, following prior work (Park et al.,

---

**Algorithm 1 Value Flows** is an RL algorithm using flow matching to model the return distribution.

---

1: **Input** The return vector field $v_\theta$, the target return vector field $v_{\bar\theta}$, the BC flow policy $\pi_\omega$, the one-step flow policy $\pi_\eta$, and the dataset $D$.
2: **for** each iteration **do**
3:     Sample a batch of transitions $\{(s, a, s', r)\} \sim \mathcal{D}$ and a batch of noises $\{\epsilon\} \sim \mathcal{N}(0, 1)$
4:     Compute the confidence weight $w(s, a, \epsilon)$ using the Euler method and VJP     ▷ Sec. 4.3
5:     Train the return vector field $v_\theta$ by minimizing $\mathcal{L}_{\text{Value Flow}}(\theta)$     ▷ Appendix C.1
    ▽ Offline RL
6:     Train the BC flow policy $\pi_\omega$ by minimizing $\mathcal{L}_{\text{BC Flow}}(\omega)$     ▷ Appendix C.2
    ▽ Online fine-tuning in offline-to-online RL
7:     Train the one-step flow policy $\pi_\eta$ by minimizing $\mathcal{L}_{\text{One-step Flow}}(\eta)$     ▷ Appendix C.2
8:     Update the target return vector field $v_{\bar\theta}$ using Polyak averages
9: **Return** $v_\theta$, $\pi_\omega$, and $\pi_\eta$.

---

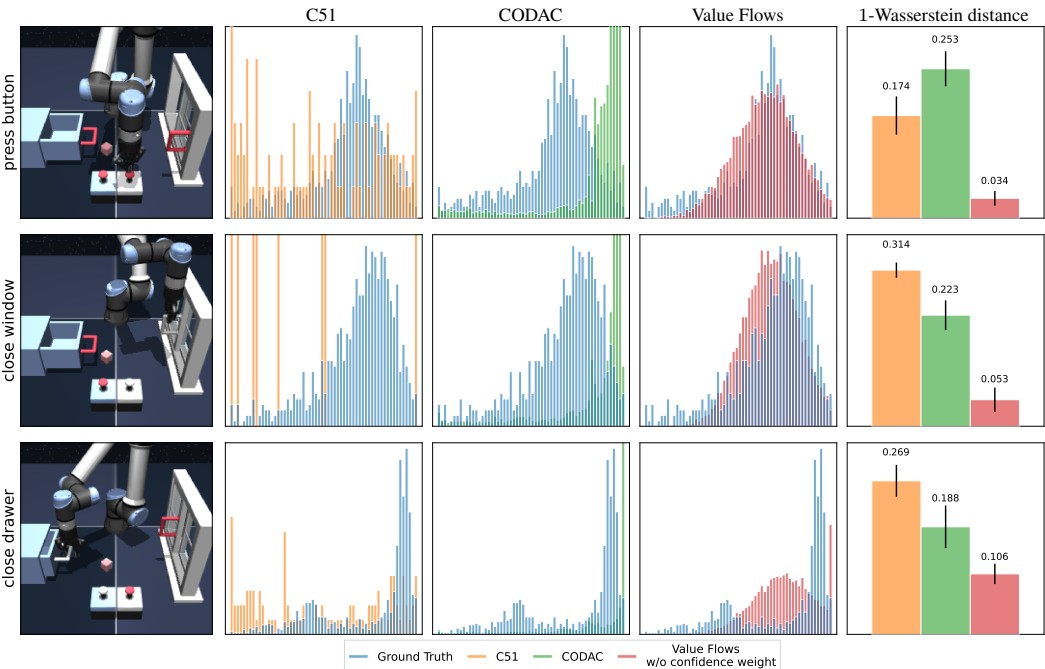

Figure 2: **Visualizing the return distribution.** *(Column 1)* The policy completes the task of closing the window and closing the drawer using the buttons to lock and unlock them. *(Column 2)* C51 predicts a noisy multi-modal distribution and *(Column 3)* CODAC collapses to a single return mode. *(Column 4)* Value Flows infers a smooth return histogram resembling the ground-truth return distribution. *(Column 5)* Quantitatively, Value Flows achieves $3\times$ lower 1-Wasserstein distance than alternative methods. See Sec. 5.1 for details.

2025c), we learn a stochastic one-step policy to maximize the Q estimates while distilling it toward the fixed BC flow policy. See Appendix C.2 for detailed discussions.

Our algorithm, Value Flows, consists of three main components: *(1)* the vector field estimating the return distribution, *(2)* the confidence weight prioritizing learning the return distribution at uncertain transition, and *(3)* the flow policy selecting actions. Using neural networks to parameterize the return vector field $v_\theta$, the BC flow policy $\pi_\omega$, and the one-step flow policy $\pi_\eta$, we optimize them using stochastic gradient descent. Alg. 1 summarizes the pseudocode of Value Flows, and our open-source implementation is available online.[3]

## 5 EXPERIMENTS

The goal of our experiments is to answer the following key questions:

(**Q1**) Does Value Flows predict returns that align with the underlying return distribution?

(**Q2**) Can Value Flows effectively learn a policy from a static offline dataset?

(**Q3**) How sample efficient is Value Flows in offline-to-online learning compared to prior methods?

(**Q4**) What are the key components of Value Flows?

**Experiment Setup.** The most relevant prior work is distributional RL methods, which model the return distribution using a categorical distribution (C51 (Bellemare et al., 2017)) or a finite number of quantiles (IQN (Dabney et al., 2018) and CODAC (Ma et al., 2021)). We will compare Value Flows against these prior methods by visualizing the return predictions and measuring the success rates in offline and offline-to-online settings. We also compare Value Flows with prior RL methods that learn the scalar Q values with a Gaussian policy (BC, IQL (Kostrikov et al., 2021a), and ReBRAC (Tarasov et al., 2023)) or a flow policy (FBRAC, IFQL, FQL (Park et al., 2025c)) for completeness. Our experiments will use standard benchmarks introduced by prior work on offline RL. We choose a set of 36 state-based tasks from OGBench (Park et al., 2025a) and D4RL (Fu et al., 2021) and choose a set of 25 image-based tasks from OGBench to evaluate our algorithm in the offline setting. Following prior work (Park et al., 2025c), we report means and standard deviations over 8 random seeds for state-based tasks (4 seeds for image-based tasks). We defer the detailed discussions about environments and datasets to Appendix E.1 and baselines to Appendix E.2. We present the hyperparameters in Appendix E.3.

### 5.1 VISUALIZING RETURN DISTRIBUTIONS OF VALUE FLOWS

One of the motivations of designing Value Flows is to use flow-based models to estimate the entire return distribution (Sec. 4.1). We hypothesize that the proposed method fits the underlying return distributions with lower distributional discrepancy. To investigate this hypothesis, we study the return predictions from Value Flows and compare them against two alternative distributional RL methods: C51 (Bellemare et al., 2017) and CODAC (Ma et al., 2021). On `scene-play-singletask-task2-v0` from OGBench, we use a learned FQL agent to collect 5000 optimal and suboptimal trajectories as our evaluation dataset to visualize the return distribution for different methods. These trajectories are out-of-distribution because *(1)* they are collected by a learned agent instead of the behavioral policy, and (2) they yield some combinatorial generalization behaviors (see Appendix E.1 for explanations). We visualize the 1D return histograms inferred by different algorithms, including the ground-truth return distribution for reference. For fair comparison, we use 5000 return samples and 60 bins for each histogram. For quantitative evaluations, we measure the 1-Wasserstein distances (Panaretos & Zemel, 2019) between the histogram of each method and the ground-truth histogram.

Fig. 2 shows the resulting histograms, superimposing the ground-truth return distribution, along with the 1-Wasserstein distances for different methods. Observe that Value Flows predicts a smooth return histogram resembling the ground-truth return distribution, while baselines either infer a noisy multi-modal distribution (C51) or collapse to a single return mode (CODAC). Numerically, our method achieves a 1-Wasserstein distance $3\times$ lower than the best-performing baseline. These results suggest that Value Flows can effectively estimate the multimodal return distribution.

### 5.2 COMPARING TO PRIOR OFFLINE AND OFFLINE-TO-ONLINE METHODS

**Offline RL.** Our next experiments compare Value Flows to prior offline RL methods, including those estimating the return distribution using alternative mechanisms. We compare our method against baselines on both state-based tasks and image-based tasks. Results in Table 1 aggregate over 5 tasks in each domain of OGBench and 12 tasks from D4RL, and we defer the full results to Appendix Table 3. These results show that Value Flows matches or surpasses all baselines on 9 out of 11 domains. On state-based OGBench tasks, most baselines performed similarly on the

---

[3]https://github.com/chongyi-zheng/value-flows

Table 1: **Offline evaluation on OGBench and D4RL benchmarks.** Value Flows achieves the best or near-best performance on 9 out of 11 domains. Following prior work (Park et al., 2025c), we average results over 8 seeds (4 seeds for image-based tasks) and bold values within 95% of the best performance for each domain. See Table 3 for full results.

| Domain | Gaussian Policies | | | Flow Policies | | | | | | |
| --- | --- | --- | --- | --- | --- | --- | --- | --- | --- | --- |
| | BC | IQL | ReBRAC | FBRAC | IFQL | FQL | C51 | IQN | CODAC | Value Flows |
| cube-double-play (5 tasks) | 2 ± 1 | 6 ± 2 | 12 ± 3 | 15 ± 6 | 14 ± 5 | 29 ± 6 | 2 ± 0 | 42 ± 8 | 61 ± 6 | **69 ± 4** |
| cube-triple-play (5 tasks) | 0 ± 0 | 1 ± 1 | 0 ± 0 | 0 ± 0 | 0 ± 0 | 4 ± 2 | 0 ± 0 | 6 ± 0 | 2 ± 1 | **14 ± 3** |
| puzzle-3x3-play (5 tasks) | 2 ± 1 | 9 ± 3 | 22 ± 2 | 14 ± 5 | 19 ± 1 | 30 ± 4 | 1 ± 0 | 15 ± 1 | 20 ± 5 | **87 ± 13** |
| puzzle-4x4-play (5 tasks) | 0 ± 0 | 7 ± 2 | 14 ± 3 | 13 ± 5 | **25 ± 8** | 17 ± 5 | 0 ± 0 | **27 ± 4** | 20 ± 18 | **27 ± 4** |
| scene-play (5 tasks) | 5 ± 2 | 28 ± 3 | 41 ± 7 | 45 ± 5 | 30 ± 4 | **56 ± 2** | 4 ± 1 | 40 ± 1 | **55 ± 1** | 59 ± 4 |
| visual-antmaze-medium-navigate (5 tasks) | - | 84 ± 5 | **87 ± 4** | 30 ± 3 | **87 ± 2** | 38 ± 4 | - | 74 ± 4 | - | 75 ± 10 |
| visual-antmaze-teleport-navigate (5 tasks) | - | 6 ± 4 | 4 ± 1 | 6 ± 3 | 10 ± 4 | 5 ± 2 | - | 4 ± 2 | - | **13 ± 4** |
| visual-cube-double-play (5 tasks) | - | 11 ± 6 | 1 ± 1 | 2 ± 1 | 2 ± 2 | 6 ± 1 | - | 1 ± 0 | - | **13 ± 2** |
| visual-puzzle-3x3-play (5 tasks) | - | 2 ± 3 | 20 ± 1 | 1 ± 1 | **21 ± 0** | 20 ± 1 | - | 19 ± 1 | - | **23 ± 2** |
| visual-scene-play (5 tasks) | - | 26 ± 5 | 28 ± 5 | 11 ± 1 | 21 ± 2 | **41 ± 4** | - | **41 ± 6** | - | **43 ± 7** |
| D4RL adroit (12 tasks) | 48 | 53 | **59** | 50 ± 3 | 52 ± 4 | 52 ± 3 | 48 ± 2 | 50 ± 3 | 52 ± 1 | 50 ± 2 |

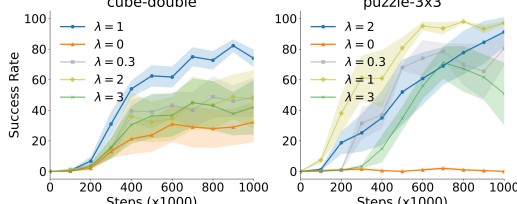

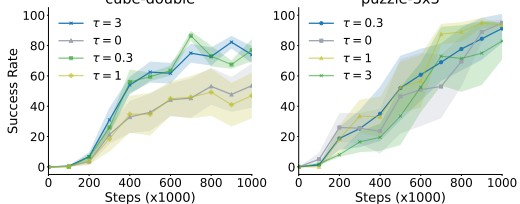

Figure 4: **Regularizing the flow-matching loss is important.** The regularization coefficient $\lambda$ needs to be tuned for better performance.

Figure 5: **Reweighting the flow-matching objective boosts success rates.** Choosing the correct confidence weight boosts the performance of Value Flows.

easier domains (`cube-double-play`, `scene-play`, and `D4RL adroit`), while Value Flows is able to obtain 40% mean improvement. On those more challenging state-based tasks, Value Flows achieves 1.6× higher success rates than the best performing baseline. In addition, Value Flows is able to outperform the best baseline by 1.24× using RGB images as input directly (visual tasks).

**Offline-to-Online RL.** We next study whether the proposed method can be used for online fine-tuning. We hypothesize that Value Flows continues outperforming prior state-of-the-art RL and distributional RL methods using online interactions, without any modifications to the distributional flow-matching losses. Results in Fig. 3 suggest that Value Flows achieves strong fine-tune performance with high sample efficiency in the online fine-tuning phase. On `puzzle-4x4-play`, Value Flows achieves 15% higher performance than all of the prior offline-to-online RL algorithms. See Appendix Table 4 for the full results and Fig. 6 for the complete learning curves. Taken together, Value Flows can be widely applied in both offline and offline-to-online settings.

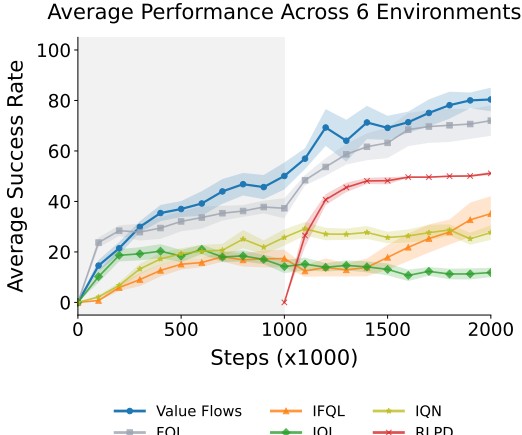

Figure 3: **Offline-to-online evaluation.** Using the same distributional flow-matching objective, Value Flows achieves higher average success rates. See Fig. 6 for the full results.

## 5.3 THE KEY COMPONENTS OF VALUE FLOWS

Our final set of experiments studies the key components of Value Flows: the regularization loss (Eq. 17), and the confidence weight (Eq. 7). To study the effects of these two components, we conduct ablation experiments on two representative state-based OGBench tasks, varying the regularization coefficient $\lambda$ for the regularization loss and the temperature $\tau$ for the confidence weight.

First, results in Fig. 4 suggest that adding the BCFM regularization loss into the distributional flow matching objective is important for making progress on the `puzzle-3x3` task, while choosing the correct regularization coefficient $\lambda$ boosts the performance of Value Flows by $2.6\times$. We also observe that the best $\lambda$ value increases the sample efficiency of our algorithm by $2\times$ compared to the unregularized variant. We conjecture that the BCFM regularization serves as an efficient anchor for learning the full return distribution. Second, Fig. 5 demonstrates that using the confidence weight to reweight the distributional flow matching loss increases the success rate, where an appropriate temperature $\tau$ results in a $60\%$ higher success rate on average. In Appendix D.1, we include additional experiments studying the robustness of Value Flows against the number of flow steps (Fig. 7) in the Euler method and the number of candidates in rejection sampling (Fig. 8) for action selections.

**Additional experiments.** In Appendix D.3, we investigate the computational time and GPU memory consumption of Value Flows. Appendix D.4 demonstrates the capability of Value Flows in an MDP requiring exploration and risk-sensitive control. In Appendix D.5, we include an ablation study showing that using Gaussian policies improves the performance of Value Flows on `D4RL adroit` tasks. The ablations in Appendix D.6 study the role of the $\mathcal{L}_{\text{BCFM}}$ loss, which mainly serves as a regularization.

## 6 CONCLUSION

We present Value Flows, an RL algorithm that uses modern, flexible flow-based models to estimate the full future return distributions. Theoretically, we show that our objective generates a probability path satisfying the distributional Bellman equation. Our experiments demonstrate that Value Flows outperforms state-of-the-art offline RL and offline-to-online RL methods in complex continuous control tasks.

**Limitations.** Although we focus on learning the full return distribution using flow-matching and reweighting our flow-matching objective using uncertainty estimation, it remains unclear how to disentangle the epistemic uncertainty from the aleatoric uncertainty with the current method. In addition, our discussion is orthogonal to the policy extraction mechanisms. Finding the appropriate policy extraction method within the distributional RL framework might reveal the true benefits of estimating the full return distribution.

## REPRODUCIBILITY STATMENT

We include additional implementation details for hyperparameters, datasets, and evaluation protocols in Appendix E. Our open-source implementations, including baselines, can be found at https://github.com/chongyi-zheng/value-flows. We run experiments for state-based tasks on A6000 GPUs for up to 6 hours, and run experiments for image-based tasks on the same type of GPUs for up to 16 hours.

## ACKNOWLEDGEMENTS

We thank Catherine Ji, Bhavya Agrawalla, and Seohong Park for their helpful discussions. We thank Seohong Park for sharing the data for some baselines in our experiments. This work used the Della computational cluster provided by Princeton Research Computing, as well as the Ionic and Neuronic computing clusters maintained by the Department of Computer Science at Princeton University. This work was in part supported by an NSF CAREER award, AFOSR YIP, ONR grant N00014-22-1-2293. This material is based upon work supported by the National Science Foundation under Award Numbers 2441665, 2237693, and 1941722. Any opinions, findings, conclusions, or recommendations expressed in this material are those of the authors and do not necessarily reflect the views of the National Science Foundation.

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

Table 2: Summary of notations and their meanings.

| Notation | Meaning |
|---|---|
| $\mathcal{S}$ | the state space |
| $\mathcal{A}$ | the action space |
| $\rho$ | the initial state distribution |
| $\Delta(\mathcal{X})$ | set of all possible probability distributions over a space $\mathcal{X}$ |
| $r(s,a)$ | the reward function |
| $\gamma$ | the discount factor |
| $p(s' \mid s,a)$ | the environmental transition distribution |
| $h$ | RL timestep |
| $F_X, p_X$ | CDF and PDF of the random variable $X$ |
| $D = \{(s,a,r,s')\}$ | the offline dataset |
| $Z^\pi, Z^\pi(s,a)$ | return random variable and conditional return random variable of policy $\pi$ |
| $Q^\pi$ | the Q function of policy $\pi$ |
| $\mathcal{T}^\pi$ | the distributional Bellman operator |
| $\epsilon \sim \mathcal{N}(0, I)$ | noise sampled from the standard Gaussian distribution |
| $\phi(\epsilon \mid t), \phi(\epsilon \mid t,s,a)$ | time-dependent diffeomorphic flow |
| $v(x^t \mid t), v(z^t \mid t,s,a)$ | time-dependent vector field |
| $p(x^t \mid t), z^t \mid t,s,a$ | time-dependent probability density path |
| $v_k(z^t \mid t,s,a), p_k(x^t \mid t,s,a)$ | return vector field and return probability density path at iteration $k$ |
| $\mathcal{L}_{\text{DFM}}$ | the distributional flow matching loss |
| $\mathcal{L}_{\text{DCFM}}$ | the distributional conditional flow matching loss |
| $\widehat{\mathbb{E}}[X], \widehat{\text{Var}}(X)$ | expectation and variance estimation of random variable $X$ |
| $w(s,a,\epsilon)$ | the confidence weight |
| $\tau$ | the confidence weight temperature |
| $\mathbb{P}(Y)$ | probability of event $Y$ |
| $\text{div}(\cdot)$ | divergence operator |
| $\mathcal{L}_{\text{CFM}}$ | the conditional flow matching loss |
| $J_v, J_\phi$ | Jacobians of vector field $v$ and flow $\phi$ |
| $\widehat{\text{Cov}}(X)$ | covariance estimation of random variable $X$ |
| $O(\cdot)$ | polynomial higher order terms |
| $\mathcal{L}_{\text{BCFM}}$ | the bootstrapped conditional flow matching loss |
| $\mathcal{L}_{\text{wDCFM}}, \mathcal{L}_{\text{wBCFM}}, \mathcal{L}_{\text{Value Flow}}$ | the weighted DCFM loss, the weighted BCFM loss, and the Value Flow loss |
| $\lambda$ | the regularization coefficient of the weighted BCFM loss |
| $\pi^\beta, \pi$ | the flow BC policy and flow one-step policy |
| $\hat{Q}(s,a)$ | Q estimation at state-action pair $(s,a)$ |
| $\mathcal{L}_{\text{One-step Flow}}$ | the flow one-step policy loss |
| $\alpha$ | the flow BC distillation coefficient in the flow one-step policy loss |

# A    PRELIMINARIES

## A.1    THE RETURN AND THE Q FUNCTION

The goal of RL is to learn a policy $\pi : \mathcal{S} \to \Delta(\mathcal{A})$ that maximizes the expected discounted return $\mathbb{E}_{\pi(S_0, A_0, S_1, A_1, \cdots)}\left[\sum_{h=0}^\infty \gamma^h r(S_h, A_h)\right]$ over trajectories $(S_0, A_0, S_1, A_1, \cdots)$. Starting from a state-action pair $(s, a)$, we denote the conditional trajectories as $(S_0 = s, A_0 = a, S_1, A_1, \cdots)$. This notation allows us to define the Q function of the policy $\pi$ as $Q^\pi(s, a) = \mathbb{E}_{\pi(S_0=s, A_0=a, S_1, A_1, \cdots)}\left[\sum_{h=0}^\infty \gamma^h r(S_h, A_h)\right]$. Prior actor-critic methods (Fujimoto et al., 2018; Haarnoja et al., 2018) typically estimate the Q function by minimizing the temporal difference (TD) error (Ernst et al., 2005; Fu et al., 2019; Fujimoto et al., 2022). We will construct TD errors to learn the entire return distribution in Sec. 4.2.

## A.2    DISTRIBUTIONAL REINFORCEMENT LEARNING

**Lemma 1** (Proposition 1 of Morimura et al. (2010)). *For a policy $\pi$, return value $z \in [z_{min}, z_{max}]$, state $s \in \mathcal{S}$, and action $a \in \mathcal{A}$, the cumulative distribution function of the random variable $\mathcal{T}^\pi Z(s,a)$,*

*i.e.,* $F_{\mathcal{T}^\pi Z}(\cdot \mid s, a)$, *satisfies*

$$F_{\mathcal{T}^\pi Z}(z \mid s, a) = \mathbb{E}_{p(s'\mid s,a),\pi(a'\mid s')}\left[F_Z\left(\left.\frac{z - r(s,a)}{\gamma}\right| s', a'\right)\right]. \tag{8}$$

For completeness, we include a full proof.

*Proof.* By the definition of identity in distribution, we have

$$F_{\mathcal{T}^\pi Z}(z \mid s, a) = F_{r(s,a)+\gamma Z(S',A')}(z \mid s, a).$$

Expanding the definition of the CDF $F_{r(s,a)+\gamma Z(S',A')}$ gives us

$$\begin{aligned}
F_{r(s,a)+\gamma Z(S',A')}(z \mid s, a) &= \mathbb{P}(r(s,a) + \gamma Z(S', A') \leq z \mid s, a) \\
&\stackrel{(a)}{=} \mathbb{E}_{p(s'\mid s,a),\pi(a'\mid s')}\left[\mathbb{P}\left(r(s,a) + \gamma Z(s', a') \leq z \mid s', a'\right)\right] \\
&= \mathbb{E}_{p(s'\mid s,a),\pi(a'\mid s')}\left[\mathbb{P}\left(\left.Z(s', a') \leq \frac{z - r(s,a)}{\gamma}\right| s', a'\right)\right] \\
&= \mathbb{E}_{p(s'\mid s,a),\pi(a'\mid s')}\left[\mathbb{P}\left(\left.Z(s', a') \leq \frac{z - r(s,a)}{\gamma}\right| s', a'\right)\right] \\
&\stackrel{(b)}{=} \mathbb{E}_{p(s'\mid s,a),\pi(a'\mid s')}\left[F_Z\left(\left.\frac{z - r(s,a)}{\gamma}\right| s', a'\right)\right]
\end{aligned}$$

in *(a)* we use the law of total probability and in *(b)* we use the definition of the CDF $F_Z$. Thus, we conclude that $F_{\mathcal{T}^\pi Z}(z \mid s, a) = \mathbb{E}_{p(s'\mid s,a),\pi(a'\mid s')}\left[F_Z\left(\left.\frac{z-r(s,a)}{\gamma}\right| s', a'\right)\right]$. $\square$

The connection between the CDFs of $\mathcal{T}^\pi Z$ and $Z$ also allows us to derive an identity between their PDFs, suggesting an alternative definition of the distributional Bellman operator:

**Lemma 2** (Chapter 4 of Bellemare et al. (2023)). *For a policy $\pi$, return value $z \in [z_{min}, z_{max}]$, state $s \in \mathcal{S}$, and action $a \in \mathcal{A}$, the probability density function of the random variable $\mathcal{T}^\pi Z(s, a)$, i.e., $p_{\mathcal{T}^\pi Z}(\cdot \mid s, a)$, satisfies*

$$p_{\mathcal{T}^\pi Z}(z \mid s, a) = \frac{1}{\gamma}\mathbb{E}_{p(s'\mid s,a),\pi(a'\mid s')}\left[p_Z\left(\left.\frac{z - r(s,a)}{\gamma}\right| s', a'\right)\right].$$

*Alternatively, the distributional Bellman operator can operate on the density function directly,*

$$\mathcal{T}^\pi p_Z(z \mid s, a) \triangleq p_{\mathcal{T}^\pi Z}(z \mid s, a). \tag{9}$$

For completeness, we include a full proof.

*Proof.* Since the PDF of a continuous random variable can be obtained by taking the derivative of its CDF, we can easily show the desired identity by taking the derivative on both sides of Eq. 8.

$$\begin{aligned}
\frac{d}{dz}F_{\mathcal{T}^\pi Z}(z \mid s, a) &= \frac{d}{dz}\mathbb{E}_{p(s'\mid s,a),\pi(a'\mid s')}\left[F_Z\left(\left.\frac{z - r(s,a)}{\gamma}\right| s', a'\right)\right], \\
p_{\mathcal{T}^\pi Z}(z \mid s, a) &\stackrel{(a)}{=} \frac{1}{\gamma}\mathbb{E}_{p(s'\mid s,a),\pi(a'\mid s')}\left[p_Z\left(\left.\frac{z - r(s,a)}{\gamma}\right| s', a'\right)\right],
\end{aligned}$$

where we use the chain rule of derivatives in *(a)*. $\square$

The stochasticity of the distributional Bellman operator $\mathcal{T}^\pi$ mainly comes from *(1)* the environmental transition $p(s' \mid s, a)$ and *(2)* the stochastic policy $\pi(a \mid s)$. We next justify the unique fixed point $Z^\pi$ of the distributional Bellman operator $\mathcal{T}^\pi$.

**Lemma 3.** *The distributional Bellman operator $\mathcal{T}^\pi$ admits the unique fixed point $Z^\pi$. Specifically, we have*

$$\begin{aligned}
\mathcal{T}^\pi Z^\pi(s, a) &\stackrel{d}{=} Z^\pi(s, a), \\
\mathcal{T}^\pi p_{Z^\pi}(z \mid s, a) &= p_{Z^\pi}(z \mid s, a).
\end{aligned} \tag{10}$$

*Proof.* By definition of $Z^\pi(s,a)$ and $\mathcal{T}^\pi$, we have

$$Z^\pi(s,a) = r(S_0 = s, A_0 = a) + \gamma \sum_{t=1}^{\infty} \gamma^{h-1} r(S_h, A_h)$$

$$\overset{d}{\underset{(a)}{=}} r(s,a) + \gamma Z^\pi$$

$$\underset{(b)}{=} r(s,a) + \gamma Z^\pi(S', A')$$

$$\overset{d}{=} \mathcal{T}^\pi Z^\pi(s,a),$$

with $S'$ and $A'$ being random variables of the next state-action pair, in *(a)* we use the stationary property of MDP, and in *(b)* we plug in the definition of $Z^\pi$. Thus, we conclude that $\mathcal{T}^\pi Z^\pi(s,a) \overset{d}{=} Z^\pi(s,a)$. By Lemma 1, the CDF of $Z^\pi(s,a)$ satisfies

$$F_{Z^\pi}(z \mid s,a) = F_{\mathcal{T}^\pi Z^\pi}(z \mid s,a) = \mathbb{E}_{p(s'|s,a),\pi(a'|s')} \left[ F_{Z^\pi}\left( \frac{z - r(s,a)}{\gamma} \,\middle|\, s', a' \right) \right].$$

Taking derivatives with respect to $z$ on both sides, the PDF of $Z^\pi(s,a)$ satisfies

$$p_{Z^\pi}(z \mid s,a) = \frac{1}{\gamma} \mathbb{E}_{p(s'|s,a),\pi(a'|s')} \left[ p_{Z^\pi}\left( \frac{z - r(s,a)}{\gamma} \,\middle|\, s', a' \right) \right].$$

We conclude that $\mathcal{T}^\pi p_{Z^\pi}(z \mid s,a) = p_{Z^\pi}(z \mid s,a)$ using the definition of the distributional Bellman operator $\mathcal{T}^\pi$ on PDFs (Lemma 2). $\qquad\square$

### A.3 FLOW MATCHING

Flow matching (Lipman et al., 2023; 2024; Liu et al., 2023; Albergo & Vanden-Eijnden, 2023) refers to a family of generative models based on ordinary differential equations (ODEs), which are close cousins of denoising diffusion models (Sohl-Dickstein et al., 2015; Song et al., 2021; Ho et al., 2020) based on stochastic differential equations (SDEs). The deterministic nature of ODEs equips flow-matching methods with simpler learning objectives and faster inference speed than denoising diffusion models (Lipman et al., 2023; 2024). The goal of flow match-

---

**Algorithm 2** Euler method for solving the flow ODE (Eq. 11).

---

1: **Input** The vector field $v$, the noise $\epsilon$, the initial flow time $t_{\text{init}}$, the final flow time $t_{\text{final}}$, and the number of steps $T$.
2: Initialize $t = t_{\text{init}}$, $\Delta t = \frac{t_{\text{final}} - t_{\text{init}}}{T}$, $x^t = \epsilon$.
3: **for** each step $t = t_{\text{init}}, t_{\text{init}} + \Delta t, \cdots, t_{\text{final}}$ **do**
4: $\quad x^{t+\Delta t} \leftarrow x^t + v(x^t \mid t) \cdot \Delta t$.
5: **Return** $x^{t_{\text{final}}}$.

---

ing methods is to transform a simple noise distribution into a target distribution over some space $\mathcal{X} \subset \mathbb{R}^d$. We will use $t$ to denote the flow time step and sample the noise $\epsilon$ from a standard Gaussian distribution $\mathcal{N}(0, I_d)$ throughout our discussions. We will use $X$ to denote the $d$-dimensional random variable that generates the target data samples using the distribution $p_X$.

Specifically, flow matching uses a time-dependent vector field $v : [0,1] \times \mathbb{R}^d \to \mathbb{R}^d$ to construct a time-dependent diffeomorphic flow $\phi : [0,1] \times \mathbb{R}^d \to \mathbb{R}^d$ (Lipman et al., 2023; 2024) that realizes the transformation from a single noise $\epsilon$ to a generative sample $\hat{x}$ by following the ODE

$$\frac{d}{dt}\phi(\epsilon \mid t) = v(\phi(\epsilon \mid t) \mid t), \quad \phi(\epsilon \mid 0) = \epsilon, \quad \phi(\epsilon \mid 1) = \hat{x}. \tag{11}$$

The vector field $v$ *generates* a time-dependent probability density path $p : [0,1] \times \mathbb{R}^d \to \Delta(\mathbb{R}^d)$ if it satisfies the continuity equation (Lipman et al., 2023)

$$\frac{\partial}{\partial t} p(x^t \mid t) + \text{div}(p(x^t \mid t) v(x^t \mid t)) = 0, \tag{12}$$

where $\text{div}(\cdot)$ denotes the divergence operator. Prior work has proposed various formulations for learning the vector field (Lipman et al., 2023; Liu et al., 2023; Albergo & Vanden-Eijnden, 2023). We adopt the simplest flow-matching objective, based on optimal transport (Liu et al., 2023), called conditional flow matching (CFM) (Lipman et al., 2023). Using $\text{UNIF}([0,1])$ to denote the uniform

distribution over the unit interval and $x^t = tx + (1-t)\epsilon$ to denote a linear interpolation between the ground truth sample $x$ and the Gaussian noise $\epsilon$, the CFM loss can be written as

$$\mathcal{L}_{\text{CFM}}(v) = \mathbb{E}_{\substack{t \sim \text{UNIF}([0,1]), \\ x \sim p_X(x), \epsilon \sim \mathcal{N}(0,I)}} \left[ \|v(x^t \mid t) - (x - \epsilon)\|_2^2 \right]. \tag{13}$$

Practically, we can generate a sample from the vector field $v$ by numerically solving the ODE (Eq. 11). We will use the Euler method (Alg. 2) as our ODE solver following prior practice (Liu et al., 2023; Park et al., 2025c).

## B THEORETICAL ANALYSIS

### B.1 ESTIMATING THE RETURN DISTRIBUTION USING FLOW-MATCHING

**Proposition 1.** *For a policy $\pi$, an iteration $k \in \mathbb{N}$, a return value $z^t \in [z_{min}, z_{max}]$, a flow time $t \in [0,1]$, a state $s \in \mathcal{S}$, a action $a \in \mathcal{A}$, and a vector field $v_k(z^t \mid t, s, a)$ that generates the probability density path $p_k(z^t \mid t, s, a)$, the new vector field $v_{k+1}(z^t \mid t, s, a)$ generates the new probability density path $p_{k+1}(z^t \mid t, s, a)$.*

*Proof.* By definition, a vector field generates a probability density path, meaning that they both satisfy the continuity equation (Eq. 12) (Lipman et al., 2023). We will check the continuity equation for $v_{k+1}(z^t \mid t, s, a)$ and $p_{k+1}(z^t \mid t, s, a)$. On one hand, for $p_{k+1}$, we have

$$\frac{\partial}{\partial t} p_{k+1}(z^t \mid t, s, a)$$

$$\overset{(a)}{=} \frac{\partial}{\partial t} \frac{1}{\gamma} \mathbb{E}_{p(s'|s,a),\pi(a'|s')} \left[ p_k \left( \frac{z^t - r(s,a)}{\gamma} \middle| t, s', a' \right) \right]$$

$$\overset{(b)}{=} \frac{1}{\gamma} \mathbb{E}_{p(s'|s,a),\pi(a'|s')} \left[ \frac{\partial}{\partial t} p_k \left( \frac{z^t - r(s,a)}{\gamma} \middle| t, s', a' \right) \right]$$

$$\overset{(c)}{=} -\frac{1}{\gamma} \mathbb{E}_{p(s'|s,a),\pi(a'|s')} \left[ \text{div} \left( p_k \left( \frac{z^t - r(s,a)}{\gamma} \middle| t, s', a' \right) v_k \left( \frac{z^t - r(s,a)}{\gamma} \middle| t, s', a' \right) \right) \right],$$

in *(a)*, we plug in the definition of $p_{k+1}(z^t \mid t, s, a)$, in *(b)*, we swap the partial differentiation with the expectation because the integral does not depend on time $t$, and in *(c)*, we apply the continuity equation for $p_k$. On the other hand, by the definition of $v_{k+1}$, we have

$$p_{k+1}(z^t \mid t, s, a) v_{k+1}(z^t \mid t, s, a)$$

$$= \frac{1}{\gamma} \mathbb{E}_{p(s'|s,a),\pi(a'|s')} \left[ p_k \left( \frac{z^t - r(s,a)}{\gamma} \middle| t, s, a \right) v_k \left( \frac{z^t - r(s,a)}{\gamma} \middle| t, s, a \right) \right].$$

Since the divergence $\text{div}(\cdot)$ is a linear operator (Kaplunovsky, 2016) and the expectation does not depend on the return value $z$, we can swap the divergence with the expectation, resulting in

$$\frac{\partial}{\partial t} p_{k+1}(z^t \mid t, s, a) = -\text{div} \left( p_{k+1}(z^t \mid t, s, a) v_{k+1}(z^t \mid t, s, a) \right), \tag{14}$$

which means $v_{k+1}$ and $p_{k+1}$ follow the continuity equation exactly. $\qquad\square$

**Lemma 4.** *Given a vector field $v_k$, the distributional flow matching loss (Eq. 4) has the minimizer $\arg\min_v \mathcal{L}_{DFM}(v, v_k) = v_{k+1}$.*

*Proof.* Since $\mathcal{L}_{\text{DFM}}$ is an MSE loss, intuitively, the minimizer of it will be $v_{k+1}$. Formally, we compute the minimizer of $\mathcal{L}_{\text{DFM}}$ by using the calculus of variations (Gelfand et al., 2000) and deriving the functional derivative of $\mathcal{L}_{\text{DFM}}$ with respect to $v$, i.e. $\delta \mathcal{L}_{\text{DFM}}(v, v_k)/\delta v$:

$$\frac{\delta \mathcal{L}_{\text{DFM}}(v, v_k)}{\delta v} = 2D(s, a, r) p_{k+1}(z^t \mid t, s, a) \left( v(z^t \mid t, s, a) - v_{k+1}(z^t \mid t, s, a) \right),$$

Setting this functional derivative to zero, we have $\arg\min_v \mathcal{L}_{\text{DFM}}(v, v_k) = v_{k+1}$. $\qquad\square$

**Lemma 5.** *Given a vector field $v_k$, the distributional conditional flow matching (Eq. 5) can be rewritten as*

$$\mathcal{L}_{DCFM}(v, v_k) = \mathbb{E}_{\substack{(s,a,r,s')\sim D, t\sim \text{UNIF}([0,1]) \\ a'\sim\pi(a'|s'), z^t\sim p_k\left(z^t|t,s',a'\right)}} \left[\left(v(r+\gamma z^t \mid t,s,a) - v_k\left(z^t\mid t,s',a'\right)\right)^2\right]$$

*Proof.* With slight abuse of notations, we use $\tilde{z}^t$ to denote the samples from the convolved probability density path $p_k((\tilde{z}^t - r)/\gamma \mid t, s, a)/\gamma$. Setting $(\tilde{z}^t - r)/\gamma = z^t$, we have

$$\mathcal{L}_{\text{DCFM}}(v, v_k) = \mathbb{E}_{\substack{(s,a,r,s')\sim D, t\sim \text{UNIF}([0,1]) \\ a'\sim\pi(a'|s'), \tilde{z}^t\sim\frac{1}{\gamma}p_k\left(\frac{\tilde{z}^t-r}{\gamma}\middle|t,s',a'\right)}} \left[\left(v(\tilde{z}^t \mid t,s,a) - v_k\left(\frac{\tilde{z}^t-r}{\gamma}\middle|t,s',a'\right)\right)^2\right]$$

$$= \mathbb{E}_{\substack{(s,a,r,s')\sim D, t\sim \text{UNIF}([0,1]) \\ a'\sim\pi(a'|s'), z^t\sim p_k\left(z^t|t,s',a'\right)}} \left[\left(v(r+\gamma z^t \mid t,s,a) - v_k\left(z^t\mid t,s',a'\right)\right)^2\right]. \quad (15)$$

$\square$

**Proposition 2.** *For a policy $\pi$, a vector field $v_k$ that generates the probability density path $p_k$ at iteration $k \in \mathbb{N}$, and a candidate vector field $v$, we have $\mathcal{L}_{DFM}(v, v_k) = \mathcal{L}_{DCFM}(v, v_k) + const.$, where the constant is independent of $v$. Therefore, the gradient $\nabla_v\mathcal{L}_{DFM}(v, v_k)$ is the same as the gradient $\nabla_v\mathcal{L}_{CDFM}(v, v_k)$.*

*Proof.* We first expand the quadratic terms in $\mathcal{L}_{\text{DFM}}$,

$$\left(v(t, z^t \mid s, a) - v_k\left(t, \frac{z^t-r}{\gamma}\middle| s', a'\right)\right)^2$$

$$= v(t, z^t \mid s, a)^2 - 2v(t, z^t \mid s, a)v_k\left(t, \frac{z^t-r}{\gamma}\middle| s', a'\right) + v_k\left(t, \frac{z^t-r}{\gamma}\middle| s', a'\right)^2$$

Since only the first two terms depend on the vector field $v$, we next examine the expectation of them respectively.

$$\mathbb{E}_{\substack{(s,a,r)\sim D, t\sim\text{UNIF}([0,1]), \\ z^t\sim p_{k+1}(z^t|t,s,a)}} \left[v(z^t \mid t,s,a)\right] \overset{(a)}{=} \mathbb{E}_{\substack{(s,a,r,s')\sim D, t\sim\text{UNIF}([0,1]), \\ a'\sim\pi(a'|s'), z^t\sim\frac{1}{\gamma}p_k\left(\frac{z^t-r}{\gamma}\middle|t,s',a'\right)}} \left[v(z^t \mid t,s',a')\right]$$

$$\mathbb{E}_{\substack{(s,a,r)\sim D, t\sim\text{UNIF}([0,1]), \\ z^t\sim p_{k+1}(z^t|t,s,a)}} \left[v(z^t \mid t,s,a)v_{k+1}(z^t \mid t,s,a)\right]$$

$$\overset{(b)}{=} \mathbb{E}_{\substack{(s,a,r)\sim D, t\sim\text{UNIF}([0,1]), \\ z^t\sim p_{k+1}(z^t|t,s,a)}} \left[v(z^t \mid t,s,a) \cdot \frac{\frac{1}{\gamma}\mathbb{E}_{p(s'|s,a),\pi(a'|s')}\left[p_k\left(\frac{z^t-r}{\gamma}\middle|t,s',a'\right)v_k\left(\frac{z^t-r}{\gamma}\middle|t,s',a'\right)\right]}{p_{k+1}(z^t \mid t,s,a)}\right]$$

$$\overset{(c)}{=} \mathbb{E}_{\substack{(s,a,r,s')\sim D, t\sim\text{UNIF}([0,1]), \\ a'\sim\pi(a'|s'), z^t\sim\frac{1}{\gamma}p_k\left(\frac{z^t-r}{\gamma}\middle|t,s',a'\right)}} \left[v(z^t \mid t,s,a)v_k\left(\frac{z^t-r}{\gamma}\middle|t,s',a'\right)\right],$$

in *(a)*, we plug in the definition of $p_{k+1}$ and use the next state $s'$ in the transition from the dataset as the sample from the transition probability, in *(b)*, we plug in the definition of $v_{k+1}$, and, in *(c)*, we use the linearity of expectation. Thus, we conclude that $\mathcal{L}_{\text{DFM}}(v, v_k) = \mathcal{L}_{\text{DCFM}}(v, v_k) + const.$, where the constant is independent of $v$. $\square$

### B.2 HARNESSING UNCERTAINTY IN THE RETURN ESTIMATION

With slight abuse of notations, we prove the following Lemmas as a generalization of Proposition 3 and Proposition 4 to the standard flow-matching problem. We will use the notations introduced in Appendix A.3.

**Lemma 6.** *(Formal statement of conclusions from Frans et al. (2025)) For a vector field $v$ learned by the conditional flow matching loss $\mathcal{L}_{CFM}$ (Eq. 13) for fitting data from the random variable $X$, noises $\epsilon$ sampled from $\mathcal{N}(0, I_d)$, we have the vector field at flow time $t = 0$ produces an estimate for the expectation $\mathbb{E}[X]$:*

$$\widehat{\mathbb{E}}[X] = \mathbb{E}_{\epsilon \sim \mathcal{N}(0, I_d)}[v(\epsilon \mid 0)].$$

*Proof.* We note that $\mathcal{L}_{CFM}$ is also an MSE loss, and will use the same idea as in Lemma 4 to find its minimizer. Specifically, we use the calculus of variations and derive the functional derivative of $\mathcal{L}_{CFM}$ with respect to $v$, i.e., $\delta\mathcal{L}_{CFM}(v)/\delta v$:

$$\frac{\delta\mathcal{L}_{CFM}(v)}{\delta v} = 2\mathbb{E}_{\substack{x \sim p_X(x), \epsilon \sim \mathcal{N}(0, I_d) \\ t \sim \text{UNIF}([0,1]), x^t = tx + (1-t)\epsilon}}\left[v(x^t \mid t) - (x - \epsilon)\right],$$

$$= 2v(x^t \mid t) - 2\mathbb{E}_{(x, \epsilon) \sim p(x, \epsilon \mid x^t, t)}[x - \epsilon]$$

Setting this functional derivative to zero, we have

$$v^\star(x^t \mid t) = \arg\min \mathcal{L}_{CFM}(v) = \mathbb{E}_{(x, \epsilon) \sim p(x, \epsilon \mid x^t, t)}[x - \epsilon] = \mathbb{E}[x - \epsilon \mid x^t, t].$$

At flow time $t = 0$, we have in $x^0 = (1 - 1) \cdot x + 1 \cdot \epsilon = \epsilon$. Since $x \sim p_X(x)$ and $\epsilon \sim \mathcal{N}(0, I_d)$ are sampled independently, the conditional distribution $p(x, \epsilon \mid x^0, 0)$ reduces to $p(x, \epsilon \mid x^0, 0) = p_X(x)$. Plugging this specific conditional distribution into the minimizer $v^\star$, we have

$$v^\star(\epsilon \mid 0) = \mathbb{E}_{x \sim p_X(x)}[x - \epsilon] = \mathbb{E}_{x \sim p_X(x)}[x] - \epsilon.$$

Taking expectation over $\epsilon \sim \mathcal{N}(0, I_d)$ gives us

$$\mathbb{E}_{\epsilon \sim \mathcal{N}(0, I_d)}[v^\star(\epsilon \mid 0)] = \mathbb{E}_{x \sim p_X(x)}[x] = \mathbb{E}[X].$$

Thus, we conclude that, for a learned vector field $v$, $\mathbb{E}_{\epsilon \sim \mathcal{N}(0, I_d)}[v(\epsilon \mid 0)]$ is an estimate for the expectation $\mathbb{E}[X]$. $\qquad\square$

We next discuss the variance estimation using a learned flow. We will use $J_v \in \mathbb{R}^{d \times d}$ to denote the Jacobian of a vector field $v(x^t \mid t)$ with respect to the input $x^t$ and $J_\phi \in \mathbb{R}^{d \times d}$ to denote the Jacobian of the corresponding diffeomorphic flow $\phi(\epsilon \mid t)$ with respect to the input $\epsilon$:

$$J_v = \begin{bmatrix} \frac{\partial v_1}{\partial x_1^t} & \cdots & \frac{\partial v_1}{\partial x_d^t} \\ \vdots & \ddots & \vdots \\ \frac{\partial v_d}{\partial x_1^t} & \cdots & \frac{\partial v_d}{\partial x_d^t} \end{bmatrix}, \quad J_\phi = \begin{bmatrix} \frac{\partial \phi_1}{\partial \epsilon_1^t} & \cdots & \frac{\partial \phi_1}{\partial \epsilon_d^t} \\ \vdots & \ddots & \vdots \\ \frac{\partial \phi_d}{\partial \epsilon_1^t} & \cdots & \frac{\partial \phi_d}{\partial \epsilon_d^t} \end{bmatrix}.$$

**Lemma 7.** *For a vector field $v$ fitting data from the random variable $X$ with the corresponding diffeomorphic flow $\phi$, two independent noises $\epsilon_0$ and $\epsilon$ sampled from $\mathcal{N}(0, I_d)$, the first-order Taylor approximation of the flow $\phi(\epsilon_0 \mid 1)$ around $\epsilon$ is*

$$\phi(\epsilon_0 \mid 1) \approx \phi(\epsilon \mid 1) + J_\phi(\epsilon \mid 1)(\epsilon_0 - \epsilon).$$

*This first-order Taylor approximation produces an estimate for the covariance Cov(X):*

$$\widehat{Cov}(X) = \mathbb{E}_{\epsilon \sim \mathcal{N}(0, I_d)}\left[J_\phi(\epsilon \mid 1)J_\phi(\epsilon \mid 1)^\top\right].$$

*Proof.* The Taylor expansion of $\phi(\epsilon_0 \mid 1)$ around $\phi(\epsilon \mid 1)$ is

$$\phi(\epsilon_0 \mid 1) = \phi(\epsilon \mid 1) + J_\phi(\epsilon \mid 1)(\epsilon_0 - \epsilon) + O(\|\epsilon_0 - \epsilon\|^2),$$

where $O(\|\epsilon_0 - \epsilon\|^2)$ denotes terms with order higher than $(\epsilon_0 - \epsilon)^\top(\epsilon_0 - \epsilon)$. Thus, we can approximate samples from the random variable $X$ using the first-order Taylor expansion of $\phi(\epsilon_0 \mid 1)$ around $\phi(\epsilon \mid 1)$,

$$\hat{x} = \phi(\epsilon_0 \mid 1) \approx \phi(\epsilon \mid 1) + J_\phi(\epsilon \mid 1)(\epsilon_0 - \epsilon).$$

By the property (affine transformation) of covariance, we have the covariance estimate for $X$ at $\epsilon$:

$$
\begin{aligned}
\widehat{\mathrm{Cov}}(X \mid \epsilon) &= \mathrm{Cov}(\phi(\epsilon_0 \mid 1)) \\
&= J_\phi(\epsilon \mid 1)\mathrm{Cov}(\epsilon_0)J_\phi(\epsilon \mid 1)^\top \\
&= J_\phi(\epsilon \mid 1)J_\phi(\epsilon \mid 1)^\top.
\end{aligned}
$$

Taking expectation over $\epsilon \sim \mathcal{N}(0, I_d)$ on both side, we conclude that

$$
\begin{aligned}
\widehat{\mathrm{Cov}}(X) &= \mathbb{E}_{\epsilon \sim \mathcal{N}(0, I_d)}\left[\widehat{\mathrm{Cov}}(X \mid \epsilon)\right] \\
&= \mathbb{E}_{\epsilon \sim \mathcal{N}(0, I_d)}\left[J_\phi(\epsilon \mid 1)J_\phi(\epsilon \mid 1)^\top\right].
\end{aligned}
$$

$\square$

We are now ready to prove Proposition 3 for the return random variable $Z^\pi(s, a)$.

**Proposition 3.** *For a policy $\pi$, a state $s \in \mathcal{S}$, a action $a \in \mathcal{A}$, two independent noises $\epsilon_0$ and $\epsilon$ sampled from $\mathcal{N}(0, 1)$, the learned return vector field $v$ fitting the conditional return distribution $Z^\pi(s, a)$, the first-order Taylor approximation of the corresponding diffeomorphic flow $\phi(\epsilon_0 \mid 1, s, a)$ around $\epsilon$ is*

$$
\phi(\epsilon_0 \mid 1, s, a) \approx \phi(\epsilon \mid 1, s, a) + J_\phi(\epsilon \mid 1, s, a)(\epsilon_0 - \epsilon).
$$

*We have the vector field at flow time $t = 0$ produces an estimate for the return expectation $\mathbb{E}\left[Z^\pi(s, a)\right]$, while the first-order Taylor approximation of the flow produces an estimate for the return variance $Var(Z^\pi(s, a))$:*

$$
\widehat{\mathbb{E}}\left[Z^\pi(s, a)\right] = \mathbb{E}_{\epsilon \sim \mathcal{N}(0, 1)}[v(\epsilon \mid 0, s, a)], \quad \widehat{Var}(Z^\pi(s, a)) = \mathbb{E}_{\epsilon \sim \mathcal{N}(0, 1)}\left[\left(\frac{\partial \phi}{\partial \epsilon}(\epsilon \mid 1, s, a)\right)^2\right].
$$

*Proof.* Applying Lemma 6 and Lemma 7 to the 1-dimensional conditional return random variable $Z^\pi(s, a)$, we get the desired estimates. $\square$

We next discuss a lemma that relates the Jacobian of the flow $J_\phi$ to the Jacobian of the vector field $J_v$ using a *flow Jacobian ODE*.

**Lemma 8.** *For a learned vector field fitting data from the random variable $X$ with the corresponding diffeomorphic flow $\phi$, a noise $\epsilon$ sampled from $\mathcal{N}(0, I_d)$, the flow Jacobian $J_\phi$ and the vector field Jacobian $J_v$ satisfy the following flow Jacobian ODE,*

$$
\frac{d}{dt}J_\phi(\epsilon \mid t) = J_v(x^t \mid t)J_\phi(\epsilon \mid t), \quad J_\phi(\epsilon \mid 0) = I_d, \tag{16}
$$

*where $x^t = \phi(\epsilon \mid t)$ follows the flow ODE (Eq. 11).*

*Proof.* By definition, the vector field $v$ and the diffeomorphic flow $\phi$ satisfy the flow ODE (Eq. 11) for any flow time $t \in [0, 1]$,

$$
\frac{d}{dt}\phi(\epsilon \mid t) = v(\phi(\epsilon \mid t) \mid t), \quad x^t = \phi(\epsilon \mid t).
$$

Taking Jacobians with respect to the $d$-dimensional noise $\epsilon$ on both sides gives us

$$
\frac{d}{dt}J_\phi(\epsilon \mid t) = J_v(x^t \mid t)J_\phi(\epsilon \mid t).
$$

Since the covariance of the noise $\epsilon$ is the identity matrix $I_d$, we set $J_\phi(\epsilon \mid 0) = I_d$ to conclude the proof. $\square$

Similarly, the learned return vector field $v$ with its diffeomorphic flow $\phi$ satisfies the following *flow derivative ODE* for the conditional return random variable $Z^\pi(s, a)$.

**Proposition 4.** *For a state $s \in \mathcal{S}$, an action $a \in \mathcal{A}$, a noise $\epsilon$ sampled from $\mathcal{N}(0,1)$, and a learned return vector field $v$ with the corresponding diffeomorphic flow $\phi$, the flow derivative $\partial\phi/\partial\epsilon$ and the vector field derivative $\partial v/\partial z$ satisfy the following flow derivative ODE,*

$$\frac{d}{dt}\frac{\partial\phi}{\partial\epsilon}(\epsilon \mid t, s, a) = \frac{\partial v}{\partial z}(z^t \mid t, s, a) \cdot \frac{\partial\phi}{\partial\epsilon}(\epsilon \mid t, s, a), \quad \frac{\partial\phi}{\partial\epsilon}(\epsilon \mid 0, s, a) = 1,$$

*where $z^t = \phi(\epsilon \mid t, s, a)$ follows the flow ODE (Eq. 11).*

*Proof.* Applying Lemma 8 to the 1-dimensional conditional return random variable $Z^\pi(s, a)$, we get the desired flow derivative ODE. $\qquad\square$

## C    COMPONENTS OF THE PRACTICAL ALGORITHM

### C.1    PRACTICAL FLOW MATCHING LOSSES

Our practical loss for fitting the return distribution involves the DCFM loss and a bootstrapped regularization based on TD learning.

We will use a target vector field $\bar{v}$ (Farebrother et al., 2025b), which aggregates all the historical vector fields $\{v_k\}$, to replace the single historical vector field in $\mathcal{L}_{\text{DCFM}}$ (Eq. 15). The reasons for introducing the target vector field $\hat{v}$ are twofold. First, using a target network is a widely used technique in RL. This technique has been proven to smooth learning and enable standard TD learning (Mnih et al., 2015; Farebrother et al., 2025a). Second, a vector field minimizing the preliminary loss $\mathcal{L}_{\text{DCFM}}$ might collapse to a constant, e.g., $v = 0$, because the vector field is fitting the output of itself. However, the same problem has already existed in a family of self-supervised representation learning methods, e.g., BYOL (Grill et al., 2020), where they learn representations to predict output from the model itself. Instead of using the model from the previous iteration, they use an exponential moving average of the historical models (target model) to predict target outputs. In the experiments of Grill et al. (2020), this target network helps prevent model collapse, and this observation motivates us to use the target vector field in Value Flows. Additionally, Tian et al. (2021) provides a theoretical characterization of how BYOL avoids this failure mode. Using $z^t \sim \bar{p}(z^t \mid t, s', a')$ to denote the sampling procedure of *(1)* first sampling a noise $\epsilon \sim \mathcal{N}(0, 1)$ and *(2)* then invoking the Euler method (Alg. 2), we have

$$\mathcal{L}_{\text{DCFM}}(v) = \mathbb{E}_{\substack{(s,a,r,s')\sim D, t\sim\text{UNIF}([0,1]) \\ a'\sim\pi(a'|s'), z^t\sim\bar{p}(z^t|t,s',a')}} \left[\left(v(r + \gamma z^t \mid t, s, a) - \bar{v}\left(z^t \mid t, s', a'\right)\right)^2\right]. \quad (17)$$

In our initial experiments, naively optimizing $\mathcal{L}_{\text{DCFM}}(v)$ produced a trivial performance on some tasks. To stabilize learning, we use the return predictions at the next state-action pair $(s', a')$ and the flow time $t = 1$ to construct a bootstrapped target return and invoke the standard conditional flow matching loss (Eq. 13). The resulting loss function, called *bootstrapped conditional flow matching* (BCFM) loss, resembles the standard Bellman error. Specifically, using $z_{\text{TD}}^1 = r(s, a) + \gamma z^1$ to denote the target return, and using $z_{\text{TD}}^t = t z_{\text{TD}}^1 + (1 - t)\epsilon$ to denote a linear interpolation between $z^1$ and $\epsilon$,[4] the BCFM loss can be written as

$$\mathcal{L}_{\text{BCFM}}(v) = \mathbb{E}_{\substack{(s,a,r,s')\sim D, t\sim\text{UNIF}([0,1]) \\ a'\sim\pi(a'|s'), z^1\sim\bar{p}(z^1|1,s',a')}} \left[\left(v(z_{\text{TD}}^t \mid t, s, a) - (z_{\text{TD}}^1 - \epsilon)\right)^2\right].$$

Therefore, using $\lambda$ to denote a balancing coefficient, the regularized flow matching loss function is $\mathcal{L}_{\text{DCFM}}(v) + \lambda\mathcal{L}_{\text{BCFM}}(v)$. Our complete loss function also incorporates the confidence weight (Sec. 4.3) into the DCFM loss and the BCFM regularization:

$$\mathcal{L}_{\text{wDCFM}}(v) = \mathbb{E}_{\substack{(s,a,r,s')\sim D, t\sim\text{UNIF}([0,1]) \\ a'\sim\pi(a'|s'), z^t\sim\bar{p}(z^t|t,s',a')}} \left[w(s, a, \epsilon) \cdot \left(v(r + \gamma z^t \mid t, s, a) - v_k\left(z^t \mid t, s', a'\right)\right)^2\right].$$

$$\mathcal{L}_{\text{wBCFM}}(v) = \mathbb{E}_{\substack{(s,a,r,s')\sim D, t\sim\text{UNIF}([0,1]) \\ a'\sim\pi(a'|s'), z^1\sim\bar{p}(z^1|1,s',a')}} \left[w(s, a, \epsilon) \cdot \left(v(z_{\text{TD}}^t \mid t, s, a) - (z_{\text{TD}}^1 - \epsilon)\right)^2\right].$$

We use $\mathcal{L}_{\text{Value Flow}}(v) = \mathcal{L}_{\text{wDCFM}}(v) + \lambda\mathcal{L}_{\text{wBCFM}}(v)$ to denote the practical loss for fitting the return distribution.

---

[4]The same noise $\epsilon$ is used to sample $z^1$ and construct $z_{\text{TD}}^t$ and $z_{\text{TD}}^1 - \epsilon$.

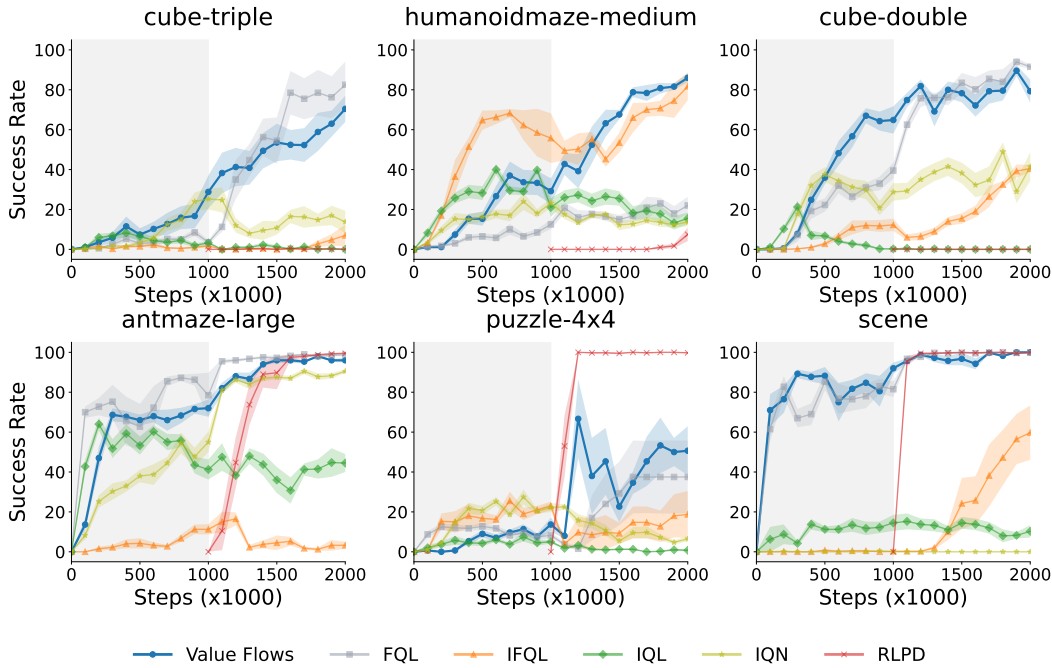

Figure 6: **Value Flows continues outperforming prior methods with online interactions.** Value Flows can be used directly for online fine-tuning and achieve strong performance without modifications to the distributional flow-matching objective. We aggregate results over 8 seeds.

### C.2 POLICY EXTRACTION STRATEGIES

We consider two different behavioral-regularized policy extraction strategies for offline RL and offline-to-online RL. First, for offline RL, following prior work (Li et al., 2025; Chen et al., 2022), we use rejection sampling to maximize Q estimates while implicitly imposing a KL constraint (Hilton, 2023) toward a fixed behavioral cloning (BC) policy. Practically, for a state $s$, we learn a stochastic BC flow policy $\pi^\beta : \mathcal{S} \times \mathbb{R}^d \to \mathcal{A}$ that transforms a $d$-dimensional noise $\epsilon_d \sim \mathcal{N}(0, I_d)$ into an action $\pi^\beta(s, \epsilon_d) \in \mathcal{A}$ using the standard conditional flow matching loss $\mathcal{L}_{\text{BC Flow}}(\pi^\beta)$ (Park et al., 2025b). Rejection sampling first uses the learned BC flow policy to sample a set of actions $\{a_1, \cdots, a_N\}$, and then selects the best action that maximizes the Q estimates (Eq. 6):

$$\hat{Q}(s, a) = \mathbb{E}_{\epsilon \sim \mathcal{N}(0,1)}[v(\epsilon \mid 0, s, a)], \quad a^\star = \underset{\{a_1, \cdots, a_N : a_i \sim \pi^\beta\}}{\arg\max} \hat{Q}(s, a_i).$$

Second, for online fine-tuning in offline-to-online RL, following prior work (Park et al., 2025c), we learn a flow one-step policy $\pi : \mathcal{S} \times \mathbb{R}^d \to \mathcal{A}$ to minimize a DDPG-style loss with behavioral regularization (Fujimoto & Gu, 2021). This loss function guides the policy to select actions that maximize the Q estimates while distilling it toward the fixed BC flow policy:

$$\mathcal{L}_{\text{One-step Flow}}(\pi) = \mathbb{E}_{s \sim D, \epsilon_d \sim \mathcal{N}(0, I_d)} \left[ -\hat{Q}(s, \pi(s, \epsilon_d)) + \alpha \|\pi(s, \epsilon_d) - \pi^\beta(s, \epsilon_d)\|^2 \right], \quad (18)$$

where $\alpha$ controls the distillation strength. The benefit of learning a parametric policy is introducing flexibility for adjusting the degree of behavioral regularization during online interactions, mitigating the issue of over-pessimism (Lee et al., 2022; Zhou et al., 2024; Nakamoto et al., 2023).

## D ADDITIONAL EXPERIMENTS

### D.1 ANALYSIS ON ABLATIONS

In this section, we conduct full ablation experiments to study the effects of *(1)* the regularization loss coefficient, *(2)* the confidence weight, *(3)* the number of flow steps in the Euler method, and *(4)* the

Table 3: **Full offline evaluation on OGBench and D4RL benchmarks.** We present the full evaluation results on 49 OGBench tasks and 12 D4RL tasks. Following prior work (Park et al., 2025c), we use an asterisk (*) to indicate the task for hyperparameter tuning in each domain. We aggregate the results over 8 seeds (4 seeds for image-based tasks), and bold values within 95% of the best performance for each task.

| | Gaussian Policies | | | Flow Policies | | | | | | |
|---|---|---|---|---|---|---|---|---|---|---|
| | BC | IQL | ReBRAC | FBRAC | IFQL | FQL | C51 | IQN | CODAC | Value Flows |
| cube-double-play-singletask-task1-v0 | 8±3 | 27±5 | 45±6 | 47±11 | 35±9 | 61±9 | 9±0 | 70±14 | 80±11 | **97±1** |
| cube-double-play-singletask-task2-v0 (*) | 0±0 | 1±1 | 7±3 | 22±12 | 9±5 | 36±6 | 0±0 | 24±9 | 63±4 | **76±7** |
| cube-double-play-singletask-task3-v0 | 0±0 | 0±0 | 4±1 | 4±2 | 8±5 | 22±5 | 0±0 | 25±6 | 66±9 | **73±4** |
| cube-double-play-singletask-task4-v0 | 0±0 | 0±0 | 1±1 | 0±1 | 1±1 | 5±2 | 0±0 | 10±1 | 13±2 | **30±5** |
| cube-double-play-singletask-task5-v0 | 0±0 | 4±3 | 4±2 | 2±2 | 17±6 | 19±10 | 0±0 | **81±8** | **82±4** | 69±5 |
| scene-play-singletask-task1-v0 | 19±6 | 94±3 | 95±2 | **96±8** | 98±3 | **100±0** | 17±3 | **100±0** | **99±0** | **99±0** |
| scene-play-singletask-task2-v0 (*) | 1±1 | 12±3 | 50±13 | 46±10 | 0±0 | 76±9 | 2±1 | 1±0 | 85±4 | **97±1** |
| scene-play-singletask-task3-v0 | 1±1 | 32±7 | 55±16 | 78±4 | 54±19 | **98±1** | 0±1 | **94±2** | 90±3 | **94±2** |
| scene-play-singletask-task4-v0 | 2±2 | 0±1 | 3±3 | 4±4 | 0±0 | 5±1 | 2±1 | 3±1 | 0±0 | 7±17 |
| scene-play-singletask-task5-v0 | **0±0** | **0±0** | **0±0** | **0±0** | **0±0** | **0±0** | **0±0** | **0±0** | **0±0** | **0±0** |
| puzzle-3x3-play-singletask-task1-v0 | 5±2 | 33±6 | 97±4 | 63±19 | 94±3 | 90±4 | 5±0 | 71±3 | 78±8 | **99±0** |
| puzzle-3x3-play-singletask-task2-v0 | 1±1 | 4±3 | 1±1 | 2±2 | 1±2 | 16±5 | 0±0 | 2±2 | 5±2 | **98±2** |
| puzzle-3x3-play-singletask-task3-v0 | 1±1 | 3±2 | 3±1 | 1±1 | 0±0 | 10±3 | 0±0 | 0±0 | 4±3 | **97±1** |
| puzzle-3x3-play-singletask-task4-v0 (*) | 1±1 | 2±1 | 2±1 | 2±2 | 0±0 | 16±5 | 0±0 | 0±0 | 5±5 | **84±24** |
| puzzle-3x3-play-singletask-task5-v0 | 1±0 | 3±2 | 5±3 | 2±2 | 0±0 | 16±3 | 0±0 | 0±0 | 6±5 | **58±39** |
| puzzle-4x4-play-singletask-task1-v0 | 1±1 | 12±2 | 26±4 | 32±9 | **49±9** | 34±8 | 0±0 | 41±2 | 37±32 | 36±3 |
| puzzle-4x4-play-singletask-task2-v0 | 0±0 | 7±4 | 12±4 | 5±3 | 4±4 | 16±5 | 0±0 | 12±4 | 10±10 | **27±5** |
| puzzle-4x4-play-singletask-task3-v0 | 0±0 | 9±3 | 15±3 | 20±10 | **50±14** | 18±5 | 0±0 | 45±7 | 33±29 | 30±4 |
| puzzle-4x4-play-singletask-task4-v0 (*) | 0±0 | 5±2 | 10±3 | 5±1 | 21±11 | 11±3 | 0±0 | 23±2 | 12±10 | **28±5** |
| puzzle-4x4-play-singletask-task5-v0 | 0±0 | 4±1 | 7±3 | 2±2 | 2±2 | 7±3 | 0±0 | **16±6** | 10±8 | 13±2 |
| cube-triple-play-singletask-task1-v0 (*) | 1±1 | 4±4 | 1±2 | 0±0 | 2±2 | 20±6 | 1±0 | 29±2 | 9±5 | **59±12** |
| cube-triple-play-singletask-task2-v0 | 0±0 | 0±0 | 0±0 | 0±0 | 0±0 | **1±2** | 0±0 | 0±0 | **1±0** | 0±0 |
| cube-triple-play-singletask-task3-v0 | 0±0 | 0±0 | 0±0 | 0±0 | 0±0 | 0±0 | 0±0 | 1±0 | 0±0 | **7±3** |
| cube-triple-play-singletask-task4-v0 | **0±0** | **0±0** | **0±0** | **0±0** | **0±0** | **0±0** | **0±0** | **0±0** | **0±0** | **0±0** |
| cube-triple-play-singletask-task5-v0 | 0±0 | 1±1 | 0±0 | 0±0 | 0±0 | 0±0 | 0±0 | 0±0 | 0±0 | 2±1 |
| pen-human-v1 | 71 | 78 | **103** | 77±7 | 71±12 | 53±6 | 69±8 | 69±3 | 67±0 | 67±9 |
| pen-cloned-v1 | 52 | 83 | **103** | 67±9 | 80±11 | 74±11 | 67±9 | 80±11 | 76±2 | 73±5 |
| pen-expert-v1 | 110 | 128 | **152** | 119±7 | 139±5 | 142±6 | 110±3 | 118±19 | 136±2 | 117±3 |
| door-human-v1 | 2 | 3 | 0 | 4±2 | **7±2** | 0±0 | 0±0 | 0±0 | 3±1 | **7±2** |
| door-cloned-v1 | 0 | **3** | 0 | 2±1 | 2±1 | 2±1 | 0±0 | 0±0 | 0±0 | 0±0 |
| door-expert-v1 | 105 | 107 | 106 | 104±1 | 104±2 | 104±1 | 104±1 | **105±0** | 104±0 | 104±1 |
| hammer-human-v1 | **3** | 2 | 0 | 2±1 | **3±1** | 1±1 | **3±1** | 2±1 | **3±1** | 1±0 |
| hammer-cloned-v1 | 1 | 2 | 5 | 2±1 | 2±1 | **11±9** | 0±0 | 0±0 | 6±0 | 1±0 |
| hammer-expert-v1 | 127 | 129 | **134** | 119±9 | 117±9 | 125±3 | 122±1 | 121±7 | 126±1 | 125±5 |
| relocate-human-v1 | **0** | **0** | **0** | **0±0** | **0±0** | **0±0** | **0±0** | **0±0** | **0±0** | **0±0** |
| relocate-cloned-v1 | 0 | 0 | **2** | 1±1 | 0±0 | 0±0 | 0±0 | 0±0 | 0±0 | 0±0 |
| relocate-expert-v1 | 108 | 106 | 108 | 105±2 | 104±3 | **107±1** | 103±0 | 103±0 | 103±2 | 105±3 |
| visual-antmaze-medium-navigate-singletask-task1-v0 (*) | - | **78±9** | 54±15 | 27±3 | **81±3** | 32±3 | - | 62±7 | - | **77±4** |
| visual-antmaze-medium-navigate-singletask-task2-v0 | - | 90±3 | **96±1** | 42±4 | 87±1 | 60±2 | - | 88±2 | - | 75±5 |
| visual-antmaze-medium-navigate-singletask-task3-v0 | - | 80±6 | **97±1** | 32±4 | 92±1 | 35±8 | - | 64±5 | - | 81±7 |
| visual-antmaze-medium-navigate-singletask-task4-v0 | - | **89±4** | **93±2** | 23±2 | 84±3 | 35±2 | - | 71±6 | - | 71±6 |
| visual-antmaze-medium-navigate-singletask-task5-v0 | - | 84±2 | **97±1** | 25±4 | 89±2 | 29±7 | - | 84±1 | - | 70±26 |
| visual-antmaze-teleport-navigate-singletask-task1-v0 (*) | - | 5±2 | 2±0 | 1±1 | 7±4 | 2±1 | - | 2±1 | - | **10±4** |
| visual-antmaze-teleport-navigate-singletask-task2-v0 | - | 10±2 | 10±3 | 6±5 | 13±3 | 6±1 | - | 7±3 | - | **17±5** |
| visual-antmaze-teleport-navigate-singletask-task3-v0 | - | 7±7 | 4±1 | 10±4 | 8±9 | 9±4 | - | 6±4 | - | **16±3** |
| visual-antmaze-teleport-navigate-singletask-task4-v0 | - | 4±6 | 4±0 | 10±2 | **18±2** | 9±1 | - | 4±2 | - | **16±5** |
| visual-antmaze-teleport-navigate-singletask-task5-v0 | - | 2±1 | 2±1 | 2±1 | 4±2 | 1±1 | - | 2±1 | - | **8±2** |
| visual-cube-double-play-singletask-task1-v0 (*) | - | **34±23** | 4±4 | 6±2 | 8±6 | 23±4 | - | 4±1 | - | **35±2** |
| visual-cube-double-play-singletask-task2-v0 | - | 3±1 | 0±0 | 2±2 | 0±0 | 0±0 | - | 0±0 | - | **4±2** |
| visual-cube-double-play-singletask-task3-v0 | - | 7±4 | 2±2 | 2±1 | 1±1 | 4±2 | - | 0±0 | - | **11±2** |
| visual-cube-double-play-singletask-task4-v0 | - | **2±1** | 0±0 | 0±0 | 0±0 | 0±0 | - | 0±0 | - | **2±1** |
| visual-cube-double-play-singletask-task5-v0 | - | 11±2 | 0±0 | 0±0 | 2±1 | 4±1 | - | 1±1 | - | **13±3** |
| visual-scene-play-singletask-task1-v0 (*) | - | 97±2 | 98±4 | 46±4 | 86±10 | 98±3 | - | 95±2 | - | **99±0** |
| visual-scene-play-singletask-task2-v0 | - | 21±16 | 30±15 | 0±0 | 0±0 | **86±8** | - | 79±15 | - | 40±27 |
| visual-scene-play-singletask-task3-v0 | - | 12±9 | 10±7 | 10±3 | 19±2 | 22±6 | - | 31±14 | - | **66±1** |
| visual-scene-play-singletask-task4-v0 | - | 1±0 | 0±0 | 0±0 | 0±0 | 1±1 | - | 0±0 | - | **10±6** |
| visual-scene-play-singletask-task5-v0 | - | **0±0** | **0±0** | **0±0** | **0±0** | **0±0** | - | **0±0** | - | **0±0** |
| visual-puzzle-3x3-play-singletask-task1-v0 (*) | - | 7±15 | 88±4 | 7±2 | **100±0** | 94±1 | - | 84±1 | - | 93±5 |
| visual-puzzle-3x3-play-singletask-task2-v0 | - | 0±0 | **12±1** | 0±0 | 0±0 | 0±0 | - | 6±2 | - | **12±1** |
| visual-puzzle-3x3-play-singletask-task3-v0 | - | 0±0 | 1±1 | 0±0 | 2±1 | 0±0 | - | 1±0 | - | **3±1** |
| visual-puzzle-3x3-play-singletask-task4-v0 | - | 1±1 | 0±1 | 0±0 | 1±0 | 5±4 | - | 3±0 | - | **6±2** |
| visual-puzzle-3x3-play-singletask-task5-v0 | - | 0±0 | 0±0 | 0±0 | 0±0 | 1±2 | - | 1±0 | - | **2±0** |

Table 4: **Offline-to-online evaluations on OGBench tasks.** We present full results on 6 OGBench tasks in the offline-to-online setting. Value Flows achieves strong fine-tuning performance compared to baselines. The results are averaged over 8 seeds, and we bold values within 95% of the best performance for each task.

| | IQN | IFQL | FQL | RLPD | IQL | Value Flows |
|---|---|---|---|---|---|---|
| antmaze-large-navigate | 55±5 → 91±1 | 11±3 → 3±2 | **87±7 → 99±12** | 0±0 → 3±2 | 41±5 → 45±5 | **72±4 → 96±1** |
| humanoidmaze-medium-navigate | 23±3 → 14±2 | **56±12 → 82±7** | 12±7 → 22±12 | 0±0 → 8±3 | 21±5 → 16±3 | **29±6 → 86±3** |
| cube-double-play | 29±4 → 42±7 | 12±3 → 41±2 | **40±11 → 92±3** | 0±0 → 0±0 | 0±0 → 0±0 | 65±7 → 79±6 |
| cube-triple-play | 25±5 → 14±6 | 2±1 → 7±5 | **4±1 → 83±12** | 0±0 → 0±0 | 3±1 → 0±0 | 29±8 → 70±7 |
| puzzle-4x4-play | 22±2 → 6±1 | 23±2 → 19±12 | 8±3 → 38±52 | **0±0 → 100±0** | 5±1 → 0±0 | 14±3 → 51±12 |
| scene-play | 0±0 → 0±0 | 0±0 → 60±14 | **82±11 → 100±1** | **0±0 → 100±0** | 15±4 → 10±3 | **92±23 → 100±0** |

number of candidates in rejection sampling. Again, we compute means and standard deviations over 8 random seeds.

To better understand the role of the BCFM regularization loss, we ablate over different values of the regularization coefficient $\lambda$ and compare the performance of Value Flows. We present the results in Fig. 4. We observe that removing the BCFM regularization loss ($\lambda = 0$) results in poor performance on both `cube-double-play-singletask-task2-v0` and `puzzle-3x3-play-singletask-task4-v0`, while Value Flows's performance starts to in-

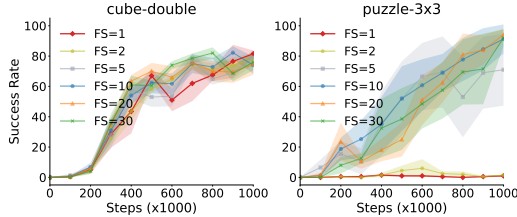 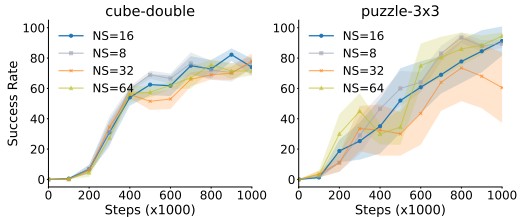

Figure 7: Value Flows is robust against the number of flow steps in the Euler method for the return vector field after a certain threshold (10).

Figure 8: The number of candidates used in rejection sampling has minor effects on the success rates of Value Flows.

crease after adding some regularizations. Choosing the correct regularization coefficient $\lambda$ boosts the mean performance of Value Flows by $2.6\times$ and increases the sample efficiency of our algorithm by $2\times$. This suggests the BCFM regularization is an important component of Value Flows.

A key design choice of Value Flows is the confidence weight. We next discuss the effect of using different temperatures $\tau$ in the confidence weight. As shown in Fig. 5, we observe that, on the `puzzle-3x3-play-singletask-task4-v0` task, using a constant confidence weight ($\tau = 0$) can result in slower convergence of the success rate. On the `cube-double-play-singletask-task2-v0` task, using a constant confidence weight results in slightly worse performance. We see that an appropriate temperature $\tau$ boosts the success rate by $60\%$ on average. Comparing against different magnitudes of the confidence weighting, we find that our choice in Table 6 is optimal.

Our final experiments study the effects of the number of flow steps in the Euler method and the number of candidates in rejection sampling. We ablate the number of flow steps within $\{1, 2, 5, 10, 20, 30\}$ and ablate the number of rejection sampling candidates within $\{8, 16, 32, 64\}$, presenting the results in Fig. 7. Note that we keep the number of flow steps for the flow policy fixed, and only vary the number of flow steps when sampling from the learned return model. We observe that, on the `cube-double-play-singletask-task2-v0` task, even using a very small number of flow steps (1 or 2) does not significantly decrease the performance of Value Flows. In contrast, using a small number of flow steps on the `puzzle-3x3-play-singletask-task4-v0` task can drastically reduce the success rate. Specifically, using 5 flow steps resulted in a $20\%$ decrease in success rate compared to the results with 10 flow steps, while using 1 or 2 flow steps will result in near-zero success rates. We also observe that the performance saturates after increasing the number of flow steps to 10. We conjecture that the number of flow steps affects the expressiveness and accuracy of the return distribution. Empirically, we found that 10 flow steps worked well and kept it fixed throughout our experiments.

We also study the effect of the number of candidates in rejection sampling for policy extraction. Results in Fig. 8 show that Value Flows is robust to the number of candidates in rejection samples, except that using 32 candidates results in a $30\%$ decrease in success rate on the `puzzle-3x3-play-singletask-task4-v0` task. In our experiments, we found that 16 candidates are sufficient across a variety of tasks.

### D.2 VISUALIZING THE CONFIDENCE WEIGHT

We visualize the confidence weight (log scale) with different temperature $\tau$ in Fig. 9, varying the input $x$ within $[0, 1]$. These results indicate that the range of our confidence weight is $[0.5, 1]$ and the confidence weight is a monotonic increasing function with respect to the input $x$. We observe that a smaller temperature $\tau$ results in a drastically increasing value in confidence weight. Given the same input $x$, a larger temperature $\tau$ results in a lower confidence weight.

### D.3 COMPUTATIONAL TIME AND RESOURCE CONSUMPTION

We analyzed our experiment logs to investigate further the computational time and GPU memory consumption of different methods. Specifically, we analyze experiment logs on one state-based OG-Bench task (`puzzle-4x4-play-singletask-task4-v0`) and one image-based OGBench

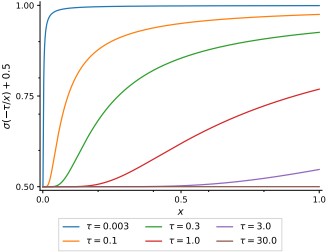

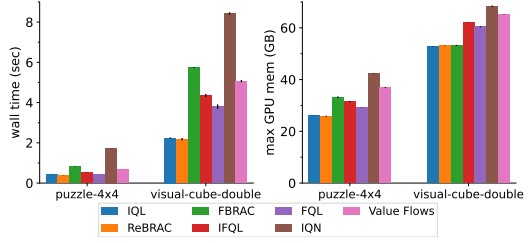

Figure 9: **The confidence weights with different temperatures.** A smaller temperature results in drastically increasing weight confidences for a larger variance in returns.

Figure 10: **Computational time and resource usage of different algorithms.** Training Value Flows requires intermediate wall time and GPU memory compared with prior methods.

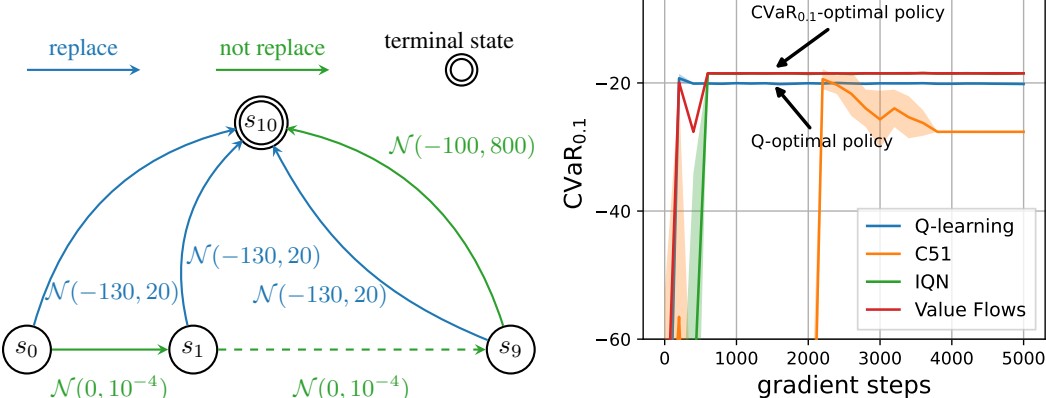

Figure 11: **The machine replacement MDP.** *(Left)* The machine replacement MDP consists of two actions *replace* and *not replace* with the reward distributed as Gaussians. This MDP is challenging because it requires exploration and risk-sensitive control. *(Right)* Value Flows converges to the optimal $\text{CVaR}_{0.1}$ with a number of gradient steps similar to IQN, while C51 failed to converge within 5000 gradient steps. The Q-learning immediately converges to the optimal expected return and is not able to learn a risk-sensitive policy.

task (`visual-cube-double-play-singletask-task1-v0`), comparing Value Flows to selection baselines: IQL, ReBRAC, FBRAC, IFQL, FQL, and IQN. We report wall-time and maximum GPU memory usage aggregated across 8 seeds (4 seeds for the image-based task) for different methods. Results in Fig. 10 suggest that training Value Flows requires a computational time similar to training FQL, and the mean wall time of Value Flows is 50% of the mean wall time of IQN. To maximize GPU usage, Value Flows used substantially less GPU memory than FBRAC and IQN because they directly use the Euler method in the actor loss (FBRAC) or employ additional neural networks to encode quantiles (IQN). Taken together, training Value Flows requires intermediate wall time and GPU memory compared with prior methods.

## D.4 SOLVING A RISK-SENSITIVE MDP

We now use a domain that requires exploration and risk-sensitive control to demonstrate the capability of Value Flows. Following prior work (Keramati et al., 2020), we choose a variant of the Machine Replacement MDP (with only 11 states) to conduct our experiments (Fig. 11 *(Left)*), measuring the $\text{CVaR}_{0.1}$ performance instead of the expected return. This MDP is especially challenging because of the chain structure, and it also requires risk-sensitive control due to the high variance in the reward distributions at the end of the chain. Therefore, it serves as a good evaluation task for distributional RL algorithms. Note that the optimal policy for expected return (Q-optimal policy) always chooses *not replace*, taking the risk of high variance reward at $s_9$. In contrast, the optimal policy for maximizing

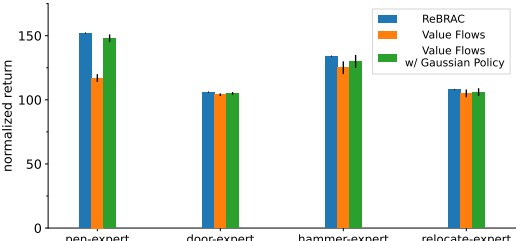
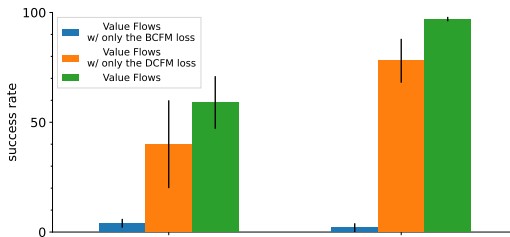

Figure 12: Using a Gaussian BC policy instead of a flow BC policy for rejection sampling improves the performance of Value Flows on D4RL adroit tasks by $10\%$, matching the state-of-the-art performance achieved by ReBRAC.

Figure 13: Training Value Flows only using the BCFM loss yields near-zero success rates. In contrast, minimizing the DCFM loss provides the most learning signal for Value Flows.

$CVaR_{0.1}$ ($CVaR_{0.1}$-optimal policy) will choose to *replace* at state $s_9$ to avoid the high variance reward.

Since the machine replacement MDP has a discrete action space, we remove the policy extraction procedure in Value Flows and use the learned vector fields to choose greedy actions directly, resembling the original C51 (Bellemare et al., 2017) and IQN (Dabney et al., 2018) implementations. We use a random policy and uniformly initialize the start state to collect a dataset with $10^5$ transitions for offline training. We compare Value Flows against C51 and IQN and use $CVaR_{0.1}$ instead of the Q-value ($CVaR_{0.5}$) to optimize all methods. We compute the means and standard deviations of $CVaR_{0.1}$ over 4 random seeds. Results in Fig. 11 *(Right)* show that Value Flows converges to the optimal $CVaR_{0.1}$ with the number of gradient steps similar to IQN (600), while C51 failed to converge within 5000 gradient steps. We also observe that Q-learning immediately converges to the optimal expected return and is not able to learn a risk-sensitive policy.

### D.5 USING GAUSSIAN POLICIES ON D4RL ADROIT TASKS

We hypothesize that flow-based policies generally performed worse than Gaussian policies on D4RL adroit tasks (see Table 1 and Table 3). This hypothesis is consistent with the experiment results in prior work, e.g., Table 2 of Park et al. (2025c). The D4RL adroit tasks consist of a very narrow dataset distribution: small amounts of human demonstrations, large amounts of expert data from a fine-tuned RL policy, and a mixture of human demonstrations and behavioral cloning trajectories (Fu et al., 2021). We conjecture that this narrow dataset distribution might impose difficulty for the expressive flow-matching policy to learn a multi-modal action distribution, resulting in sampling out-of-distribution actions during policy extraction.

To test this hypothesis, we ran additional ablations to study the effects of using Gaussian policies in Value Flows on D4RL adroit tasks. Instead of learning a flow BC policy as in Appendix C.2, we learn a Gaussian BC policy and then apply rejection sampling. We choose to conduct experiments on the four `expert` variants of D4RL adroit tasks: `pen-expert-v1`, `door-expert-v1`, `hammer-expert-v1`, and `relocate-expert-v1`, aggregating results over 8 seeds. Results in Fig. 12 suggest that using Gaussian BC policies improves the mean performance of Value Flows by $10\%$, matching the state-of-the-art performance achieved by ReBRAC.

### D.6 ROLE OF THE BOOTSTRAPPED CONDITIONAL FLOW MATCHING LOSS

In the complete Value Flows flow matching objective, we include a BCFM regularization loss $\mathcal{L}_{wBCFM}$ to stabilize learning. This section studies whether the DCFM loss $\mathcal{L}_{wDCFM}$ or the BCFM loss $\mathcal{L}_{wBCFM}$ provides the most learning signal for Value Flows. We hypothesize that simply using the BCFM loss to train the vector field in Value Flows does not provide a sufficient learning signal to model the return distribution.

To test this hypothesis, we conduct ablation experiments comparing a variant of Value Flows trained only using the BCFM loss against the original Value Flows. We evaluate the success rates on two OGBench tasks `cube-triple-play-singletask-task1-v0` and

`scene-play-singletask-task2-v0`, computing means and standard deviations of success rates after training over 8 seeds. Results in Fig. 13 suggest that training Value Flows only using the BCFM loss yields near-zero success rates on both tasks. In contrast, minimizing the DCFM loss provides the most learning signal for Value Flows. Thus, the BCFM loss mainly serves as a regularization.

# E    EXPERIMENT DETAILS

We implement Value Flows and all baselines using JAX (Bradbury et al., 2018), adapting the FQL (Park et al., 2025c) codebase. Our open-source implementations can be found at https://github.com/chongyi-zheng/value-flows.

## E.1    ENVIRONMENTS

We use OGBench (Park et al., 2025a) and D4RL (Fu et al., 2021) benchmarks to perform experimental evaluations. OGBench is a benchmark for offline goal-conditioned reinforcement learning, with single-task variants for standard reward-maximizing RL algorithms. We choose the single-task variants for all domains in Table 3. Following prior work (Park et al., 2024), we use the default task from each domain to tune hyperparameters. For locomotion tasks, the reward is either $-1$ or $0$, where $0$ indicates task completion. For manipulation tasks, the reward varies between $-n_{\text{task}}$ and $0$, depending on how many subtasks are complete. The datasets were collected by scripted non-Markovian policies that randomly interact with different objects in the environment. However, the successful completion of each task requires the agent to sequentially arrange those objects in a specific order (sequential reasoning), which is unseen in the dataset. Hence, the agent must be able to generalize across trajectories in the dataset and plan and sequentially combine learned manipulation skills (combinatorial generalization) (Park et al., 2025a). We use the following datasets from OGBench for each domain.

- `cube-double-play-v0`
- `cube-triple-play-v0`
- `scene-play-v0`
- `puzzle-3x3-play-v0`
- `puzzle-4x4-play-v0`
- `visual-antmaze-medium-navigate-v0`
- `visual-antmaze-teleport-navigate-v0`
- `visual-scene-play-v0`
- `visual-puzzle-3x3-play-v0`
- `visual-cube-double-play-v0`

In our offline-to-online experiments, we chose to conduct experiments on the following task.

- `antmaze-large-navigate-singletask-task1-v0`
- `humanoidmaze-medium-navigate-singletask-task1-v0`
- `cube-double-play-singletask-task2-v0`
- `cube-triple-play-singletask-task1-v0`
- `scene-play-singletask-task2-v0`
- `puzzle-4x4-play-singletask-task4-v0`

D4RL is a popular benchmark for studying offline RL. We measure the performance by the normalized returns following Fu et al. (2021). We use the following tasks from Adroit, which require dexterous manipulation within a 24-DoF action space.

- `pen-human-v1`

- `pen-cloned-v1`
- `pen-expert-v1`
- `door-human-v1`
- `door-cloned-v1`
- `door-expert-v1`
- `hammer-human-v1`
- `hammer-cloned-v1`
- `hammer-expert-v1`
- `relocate-human-v1`
- `relocate-cloned-v1`
- `relocate-expert-v1`

The `cube-double` and `cube-triple` tasks involve complex pick-and-place of colored cube blocks. The `scene` tasks involve long-horizon reasoning and interaction with various objects in the scene. The `puzzle-3x3` and `puzzle-4x4` tasks require solving the "Lights Out" puzzles using the robot arm and further test combinatorial generalization. The Adroit benchmarks involve controlling a 28-DoF hand to perform dexterous skills: spinning a pen, opening a door, relocating a ball, and using a hammer to knock a button. The `antmaze` and `humanoidmaze` tasks are designed to navigate a challenging maze by controlling an 8-DoF ant and a 21-DoF humanoid, respectively. The visual variants of these navigation tasks are challenging, as there is no low-dimensional state information, and the algorithm must directly learn from $64 \times 64 \times 3$ RGB images. We include challenging state-based and image-based variants for manipulation tasks as they require tackling multimodal returns and long-horizon reasoning. We also include some image-based navigation tasks to evaluate the algorithms.

### E.2 METHODS FOR COMPARISON

We compare Value Flows against 9 baselines, measuring the performance using success rates (OGBench) and discounted cumulative returns (D4RL). Specifically, we first compare to BC, IQL (Kostrikov et al., 2021a), and ReBRAC (Tarasov et al., 2023) as representative methods that learn scalar Q values with a Gaussian policy. The second set of baselines is state-of-the-art methods that learn scalar Q values with a flow policy (FBRAC, IFQL, FQL) (Park et al., 2025c). We also include comparisons against prior distributional RL methods that model the return distribution as a categorical distribution (C51 (Bellemare et al., 2017)) or a finite number of quantiles (IQN (Dabney et al., 2018), and CODAC (Ma et al., 2021)). We use a flow policy for all these distributional RL baselines. Following prior work (Park et al., 2025c), we report means and standard deviations over 8 random seeds for state-based tasks (4 seeds for image-based tasks). We describe the details of each method below.

**BC.** Behavior cloning maximizes the log likelihood of the action at a given state. We train a Gaussian policy to maximize log likelihood with a $(512, 512, 512, 512)$ MLP as the backbone.

**IQL (Kostrikov et al., 2021a).** Implicit Q-Learning is an offline RL method that uses expectile regression to learn the best action within the support of the behavioral dataset. Following prior work (Park et al., 2025c), we learn a Gaussian policy via AWR and perform hyperparameter search over the AWR inverse temperatures $\alpha$ in $\{0.3, 1, 3, 10\}$ for each environment (Table 6).

**ReBRAC (Tarasov et al., 2023).** ReBRAC is an offline actor-critic method based on TD3+BC (Fujimoto & Gu, 2021). This algorithm implements several design choices, such as layer normalization and critic decoupling for better performance. We perform sweeps over the actor and critic BC coefficients. For the actor BC coefficient $\alpha_1$, we search over values in $\{0.003, 0.01, 0.03, 0.1, 0.3, 1\}$. For the critic BC coefficient, we search over values in $\{0, 0.001, 0.01, 0.1\}$. Results for different tasks can be found in Table 6.

**IFQL (Park et al., 2025c).** IFQL is a variant of Implicit Diffusion Q-Learning (IDQL; (Hansen-Estruch et al., 2023a)) with flow policy. IFQL uses the flow policy to propose candidates and uses rejection sampling to select actions as in Value Flows. We select the number of candidates in rejection sampling $N$ from $\{32, 64, 128\}$ (Table 6).

**FBRAC (Wu et al., 2019).** Following prior work (Park et al., 2025c), we implement FBRAC as a variant of behavior-regularized actor-critic (BRAC) with flow policies. This method requires backpropagating the gradients through the ODE solver. We select the BC coefficient $\alpha$ from $\{1, 3, 10, 30, 100, 300\}$ (Table 6).

**FQL (Park et al., 2025c).** Flow Q-Learning uses a one-step flow policy to maximize the Q estimations learned by the standard TD error. It also incorporates a behavioral regularization term towards a BC flow policy (Eq. 18). We consider the BC distillation coefficient $\alpha$ in $\{3, 10, 30, 100, 300, 1000\}$ (Table 6).

**C51 (Bellemare et al., 2017).** C51 is a distributional RL method that discretizes the return distribution into a fixed number of bins and uses cross-entropy loss to update these categorical distributions. We tune the number of atoms $N$ in $\{51, 101\}$ (Table 6). Because the original C51 algorithm aims to solve the Atari game with a *discrete* action space, it greedily selects actions that maximize the Q-value estimated by the distributional returns (no explicit policy). However, since our evaluation benchmarks (Appendix E.1) have continuous action spaces, we choose to learn an explicit flow BC policy and use rejection sampling to select actions (Appendix C.2), resembling the greedy action selection in a discrete action space. Using flow-based rejection sampling for C51 not only matches the policy extraction method of Value Flows, but also prevents sampling out-of-distribution actions (Chen et al., 2023).

**IQN (Dabney et al., 2018).** IQN is a distributional RL method that approximates the return distribution by predicting quantile values at randomly sampled quantile fractions. We select the temperature $\kappa$ in the quantile regression loss from $\{0.7, 0.8, 0.9, 0.95\}$ (Table 6). Similar to the adaptation in C51, we also learn an explicit flow BC policy and use rejection sampling (Appendix C.2) to select actions that maximize the Q estimated by IQN.

**CODAC (Ma et al., 2021).** CODAC combines the IQN critic loss with conservative constraints. Similar to Park et al. (2025c), we use a one-step flow policy to perform DDPG-style policy extraction (Appendix C.2). We fixed the conservative penalty coefficient to $0.1$. We sweep the temperature $\kappa$ in the quantile regression loss over $\{0.7, 0.8, 0.9, 0.95\}$ and sweep the BC coefficient $\alpha_1$ over $\{100, 300, 1000, 3000, 10000, 30000\}$ (Table 6).

**Value Flows.** Our method, Value Flows, uses flexible flow-based models to estimate the full distribution of future returns and to identify epistemic uncertainty across different states using the learned generative flow models. We describe the components of the practical Value Flows algorithm in Appendix C.1. In our experiments, we sweep the BCFM regularization coefficient $\lambda \in \{0.3, 0.5, 1, 3, 10\}$ and the confidence weight temperature $\tau \in \{0.01, 0.03, 0.1, 0.3, 1, 3\}$. In addition, the flow-based distillation strength $\alpha$ is the key hyperparameter for our offline-to-online experiments. We select the distillation strength $\alpha$ from $\{10, 30, 100, 300\}$. See Table 6 for the specific values of the hyperparameter for each domain or task.

### E.3 HYPERPARAMETERS

We include common hyperparameters for Value Flows and baselines in Table 5 and domain-specific hyperparameters for different methods in Table 6.

---

[5]The hyperparameters for `antmaze-large-navigate` and `humanoidmaze-medium-navigate` are used for offline-to-online experiments.

Table 5: Common hyperparameters for Value Flows and baselines.

| Hyperparameter | Value |
|---|---|
| optimizer | Adam (Kingma, 2014) |
| batch size | 256 |
| learning rate | $3 \times 10^{-4}$ |
| MLP hidden layer sizes | $(512, 512, 512, 512)$ |
| MLP activation function | GELU (Hendrycks & Gimpel, 2016) |
| use actor layer normalization | Yes |
| use value layer normalization | Yes |
| number of flow steps in the Euler method | 10 |
| number of candidates in rejection sampling | 16 |
| target network update coefficient | $5 \times 10^{-3}$ |
| number of Q ensembles | 2 |
| image encoder | small IMPALA encoder (Espeholt et al., 2018; Park et al., 2025c) |
| image augmentation method | random cropping |
| image frame stack | 3 |

Table 6: **Domain specific hyperparameters for Value Flows and baselines.** The complete descriptions of each hyperparameter can be found in Appendix E.2. Following prior work Park et al. (2025c), we tune these hyperparameters for each domain from OGBench benchmarks on the default ("∗") task in Table 3. "-" indicates the values do not exist.[5]

| Domain or task | Discount $\gamma$ | IQL $\alpha$ | ReBRAC $\alpha_1$ | $\alpha_2$ | FBRAC $\alpha$ | IFQL $N$ | FQL $\alpha$ | C51 $N$ | IQN $\kappa$ | CODAC $\kappa$ | $\alpha$ | Value Flows $\lambda$ | $\tau$ | $\alpha$ |
|---|---|---|---|---|---|---|---|---|---|---|---|---|---|---|
| antmaze-large-navigate | 0.99 | 10 | - | - | - | 32 | 10 | - | 0.7 | - | - | 10 | 0.01 | 30 |
| humanoidmaze-medium-navigate | 0.995 | 10 | - | - | - | 32 | 30 | - | 0.7 | - | - | 3 | 0.3 | 100 |
| cube-double-play | 0.995 | 0.3 | 0.1 | 0 | 100 | 32 | 100 | 101 | 0.9 | 0.95 | 300 | 1 | 3 | 300 |
| cube-triple-play | 0.995 | 10 | 0.03 | 0 | 100 | 32 | 300 | 51 | 0.8 | 0.95 | 1000 | 3 | 0.03 | 300 |
| puzzle-3x3-play | 0.99 | 10 | 0.3 | 0.001 | 100 | 32 | 1000 | 101 | 0.8 | 0.95 | 100 | 0.5 | 0.3 | - |
| puzzle-4x4-play | 0.99 | 3 | 0.3 | 0.01 | 300 | 32 | 1000 | 101 | 0.95 | 0.95 | 1000 | 3 | 100 | 300 |
| scene-play | 0.99 | 10 | 0.1 | 0.001 | 100 | 32 | 300 | 51 | 0.95 | 0.95 | 100 | 1 | 0.3 | 300 |
| visual-antmaze-medium-navigate | 0.99 | 1 | 0.01 | 0.003 | 100 | 32 | 100 | - | 0.9 | - | - | 0.3 | 0.03 | - |
| visual-antmaze-teleport-navigate | 0.99 | 1 | 0.01 | 0.003 | 100 | 32 | 100 | - | 0.8 | - | - | 0.3 | 0.03 | - |
| visual-cube-double-play | 0.995 | 0.3 | 0.1 | 0 | 100 | 32 | 100 | - | 0.9 | - | - | 1 | 0.3 | - |
| visual-puzzle-3x3-play | 0.99 | 10 | 0.3 | 0.01 | 100 | 32 | 300 | - | 0.8 | - | - | 0.3 | 0.3 | - |
| visual-scene-play | 0.99 | 10 | 0.1 | 0.003 | 100 | 32 | 100 | - | 0.95 | - | - | 1 | 0.3 | - |
| pen-human | 0.99 | - | - | - | 30000 | 32 | 10000 | 51 | 0.8 | 0.8 | 10000 | 3 | 1 | - |
| pen-cloned | 0.99 | - | - | - | 10000 | 32 | 10000 | 51 | 0.8 | 0.8 | 10000 | 3 | 1 | - |
| pen-expert | 0.99 | - | - | - | 30000 | 32 | 3000 | 51 | 0.8 | 0.8 | 10000 | 3 | 0.01 | - |
| door-human | 0.99 | - | - | - | 30000 | 32 | 30000 | 101 | 0.9 | 0.9 | 10000 | 3 | 0.01 | - |
| door-cloned | 0.99 | - | - | - | 10000 | 128 | 30000 | 51 | 0.9 | 0.9 | 30000 | 10 | 0.3 | - |
| door-expert | 0.99 | - | - | - | 30000 | 32 | 30000 | 51 | 0.9 | 0.9 | 10000 | 10 | 0.3 | - |
| hammer-human | 0.99 | - | - | - | 30000 | 32 | 30000 | 51 | 0.7 | 0.8 | 30000 | 3 | 0.3 | - |
| hammer-cloned | 0.99 | - | - | - | 10000 | 32 | 10000 | 51 | 0.7 | 0.8 | 10000 | 3 | 0.3 | - |
| hammer-expert | 0.99 | - | - | - | 30000 | 32 | 30000 | 51 | 0.9 | 0.8 | 10000 | 10 | 1 | - |
| relocate-human | 0.99 | - | - | - | 30000 | 128 | 10000 | 101 | 0.9 | 0.9 | 30000 | 10 | 0.01 | - |
| relocate-cloned | 0.99 | - | - | - | 3000 | 32 | 30000 | 51 | 0.9 | 0.9 | 30000 | 3 | 0.01 | - |
| relocate-expert | 0.99 | - | - | - | 30000 | 32 | 30000 | 101 | 0.9 | 0.9 | 10000 | 3 | 0.1 | - |

