# OpenReview forum: "Value Flows"
_ICLR.cc/2026/Conference — ICLR 2026 Poster_

### Official Review · Reviewer_uJ4U · 2025-10-21

**Soundness:** 3
**Presentation:** 2
**Contribution:** 3
**Rating:** 6
**Confidence:** 2

**Summary:**

This paper proposes Value Flows, which utilizes expressive flow-based model to represent the conditional return random variable. Specifically, Value Flows learns a parameterized vector field model to estimate for the return expectation and variance for full future return distributions. Experiments on 37 state-based and 25 image-based benchmark tasks demonstrate the effectiveness of Value Flows when compared with various baselines in offline and offline-to-online settings.

**Strengths:**

1.	The motivation of this paper of using flow matching to model complex, multi-modal distributions in the domain of RL is clear. Value Flows deeply integrates flow matching into distributional RL setting to estimate return distribution.
2.	Both the theoretical and experimental parts are strong. For example, Value Flows achieves the best or near-best performance on 9 out of 11 domains.
3.	In Figure 2, they authors provide the example to show Value Flows has better 1-Wasserstein distance when modeling return distribution than both C51 and CODAC.

**Weaknesses:**

1.	The method section is hard to follow as it is full of symbols and the logic is not explicit. The authors could provide more illustrations to explain the how to connect flow matching to distributional RL.
2.	There are some typos. In Line 065, the format of citations is not correct.
3.	Some details, such as the complete loss function, are omitted in the main paper and provide in the Appendix, making the paper not self-contained.

**Questions:**

1.	Could you provide more explanations of vector field, such as its illustration in flow matching and its specific role in Value Flow?
2.	Why does Value Flows perform worse than ReBRAC and even IQL in D4RL adroit tasks?
3.	Could the authors compare the compute efficacy of Value Flows with other methods?
4.	If the flow step is very small such as 1 or 2, how could it affect the performance of Value Flows?
5.	Why is the target return vector field introduced in the algorithm of Value Flows?

---

> ### Author Response · Authors · 2025-11-24
>
> We thank the reviewer for the constructive feedback for improving the paper. The reviewer brings up three helpful suggestions: (1) a summary of the mathematical symbols and clarifications of logic, (2) the computational efficiency of Value Flows against prior methods, and (3) an ablation study for Value Flows with very small amounts of flow steps. We have attempted to incorporate a table to summarize all the notations (Table 2). To address the concern about computational efficiency, we analyze the experiment logs to compare the computation time and GPU memory consumption of different methods (Appendix D.3). We also conduct additional ablations to study the effect of using very small amounts of flow steps in Value Flows (Appendix D.1). **Together with the discussion below, does this fully address the reviewer’s questions?**
>
> > The method section is hard to follow as it is full of symbols and the logic is not explicit.
>
> To address this feedback, we include a table (Table 2) that summarizes all the mathematical notations and their meanings for reference. We have also added additional signposting at the start of the method section (Sec. 4) to explain the logical flow of the section. Please let us know if there are specific paragraphs that are still difficult to understand. We would be happy to incorporate your feedback to further improve the paper.
>
> > If the flow step is very small such as 1 or 2, how could it affect the performance of Value Flows?
>
> We incorporated the suggestion from the reviewer and ran additional ablation experiments studying the performance of Value Flows with 1 or 2 flow steps. We have revised Fig. 7 and Appendix D.1 (see the purple text) to include these new results.
>
> We choose to use the same OGBench tasks as in Fig. 7 ($\texttt{cube-double-play-singletask-task2-v0}$ and $\texttt{puzzle-3x3-play-singletask-task4-v0}$) and aggregate results over 8 random seeds. We plot the new learning curves in Fig. 7, together with other variants of the number of flow steps. These results indicate that, on the $\texttt{cube-double-play-singletask-task2-v0}$, even using a very small number of flow steps ($1$ or $2$) does not significantly decrease the performance of Value Flows. In contrast, using a small number of flow steps on the $\texttt{puzzle-3x3-play-singletask-task4-v0}$ task can drastically reduce the success rate. Specifically, using $5$ flow steps resulted in a $20\%$ decrease in success rate compared to the results with $10$ flow steps, while using $1$ or $2$ flow steps will result in near-zero success rates. We also observe that the performance saturates after increasing the number of flow steps to $10$. We conjecture that the number of flow steps affects the expressiveness and accuracy of the return distribution. Empirically, we found that $10$ flow steps worked well and kept it fixed throughout our experiments.
>
> > Could the authors compare the compute efficacy of Value Flows with other methods?
>
> As suggested by the reviewer, we analyzed our experiment logs to further investigate the computational time and GPU memory consumption of different methods. Specifically, we analyze experiment logs on one state-based OGBench task ($\texttt{puzzle-4x4-play-singletask-task4-v0}$) and one image-based OGBench task ($\texttt{visual-cube-double-play-singletask-task1-v0}$), comparing Value Flows to selection baselines in Table 3: IQL, ReBRAC, FBRAC, IFQL, FQL, and IQN. We report wall-time and maximum GPU memory usage aggregated across 8 seeds (4 seeds for the image-based task) for different methods. Results in Fig. 10 suggest that training Value Flows requires a computational time similar to training FQL, and the mean wall time of Value Flows is $50\%$ of the mean wall time of IQN. For the maximum GPU usage, Value Flows used substantially less GPU memory than FBRAC and IQN because they directly use the Euler method in the actor loss (FBRAC) or use additional neural networks to encode quantiles (IQN). Taken together, training Value Flows requires intermediate wall time and GPU memory compared with prior methods. We have added these new results to Appendix D.5 (see the purple text).

---

> > ### Author Response · Authors · 2025-11-24
> >
> > > Why does Value Flows perform worse than ReBRAC and even IQL in D4RL adroit tasks?
> >
> > We hypothesize that flow-based policies generally performed worse than Gaussian policies on D4RL adroit tasks (see Table 1). This hypothesis is consistent with the observation in prior work, e.g., Table 2 of Park et al. (2024) [1]. We note that the D4RL adroit tasks have a very narrow dataset distribution: small amounts of human demonstrations, large amounts of expert data from a fine-tuned RL policy, and a mixture of human demonstrations and behavioral cloning trajectories [2]. This narrow dataset distribution might impose difficulty for the expressive flow-matching policy to learn a multi-modal action distribution, resulting in sampling out-of-distribution actions during policy extraction.
> >
> > To test this hypothesis, we ran additional ablations to study the effects of using Gaussian policies in Value Flows on D4RL adroit tasks. Instead of learning a flow BC policy and then applying rejection sampling as in Appendix C.2, we learn a Gaussian BC policy and then apply rejection sampling. We choose to conduct experiments on the four $\texttt{expert}$ variants of D4RL adroit tasks: $\texttt{pen-expert-v1}$, $\texttt{door-expert-v1}$, $\texttt{hammer-expert-v1}$, and $\texttt{relocate-expert-v1}$, aggregating results over 8 seeds. Results in Fig. 12 suggest that using Gaussian BC policies improves the mean performance of Value Flows by $10\%$, matching the state-of-the-art performance achieved by ReBRAC.
> >
> > > Could you provide more explanations of vector field, such as its illustration in flow matching and its specific role in Value Flow?
> >
> > The goal of flow matching methods is to transform a simple noise distribution into a target distribution, i.e., generative modeling. To achieve this goal, the flow matching method uses a time-dependent vector field $v$ to construct a time-dependent diffeomorphic flow $\phi$ that realizes the transformation from a single Gaussian noise $\epsilon$ to a generative sample $\hat{\psi}$ by following the ODE
> > \begin{align*}
> >  \frac{d}{d t} \phi(\epsilon \mid t) = v(\phi(\epsilon \mid t) \mid t),\quad \phi(\epsilon \mid 0) = \epsilon,\quad \phi(\epsilon \mid 1) = \hat{x}.
> > \end{align*}
> > Intuitively, the vector field can be interpreted as the velocity of the flow, which transforms the noise into target data. We have provided detailed preliminaries of flow-matching methods in Appendix A.3 and will bring them back into the main text in the final version. Additionally, detailed discussions on flow-matching methods can be found in Lipman et al. (2022) [3] or the tutorial Lipman et al. (2024) [4].
> >
> > In the context of Value Flows, we use a vector field to model the future return distribution (Eq. 1). Since the future return distribution depends on a state-action pair $(s, a)$ and the return is a 1-dimensional scalar, we define the time-dependent return vector field as $v: \mathbb{R} \times [0, 1] \times \mathcal{S} \times \mathcal{A} \to \mathbb{R}$. We are happy to further explain the concept if there are specific paragraphs that are still difficult to understand.
> >
> > > Why is the target return vector field introduced in the algorithm of Value Flows?
> >
> > We have revised Sec. 4.2 (see the purple text) and Appendix C.1 to clarify this detail. Our vector field update rule is iterative: since computing the current vector field requires the vector field from previous iterations (Eq. 3), we need to maintain a historical vector field. Thus, we choose to use a target vector field, which is an exponential moving average over all historical vector fields (line 8 in Alg. 1) instead of a single historical vector field, in our temporal-difference (TD) flow-matching loss (Eq. 5).
> >
> > The reasons for introducing the target vector field are twofold. First, using a target network is a widely used technique in RL. This technique has been proven to smooth learning and enable standard TD learning [5, 6]. Second, as mentioned by the reviewer pDXE, a vector field minimizing the preliminary loss in Eq. 5 might collapse to a constant, e.g., $v = 0$, because the vector field is fitting the output of itself. However, the same problem has already existed in a family of self-supervised representation learning methods, e.g., BYOL [7], where they learn representations to predict output from the model itself. Instead of using the model from the previous iteration, they use an exponential moving average of the historical models (target model) to predict target outputs. In the experiments of Grill et al. (2020), this target network helps prevent model collapse, and this observation motivates us to use the target vector field in Value Flows. Taken together, using a target vector field is both a standard technique in RL and prevents collapsing.

---

> > > ### Author Response · Authors · 2025-11-24
> > >
> > > > There are some typos. In Line 065, the format of citations is not correct.
> > >
> > > Thanks for the suggestion. We have fixed the typo.
> > >
> > > > Some details, such as the complete loss function, are omitted in the main paper and provided in the Appendix, making the paper not self-contained.
> > >
> > > We will pull these details back into the main text for the camera-ready paper, which allows an additional page.
> > >
> > > [1] Park et al. (2025). Flow Q-Learning.
> > >
> > > [2] Fu et al (2020). Datasets for Deep Data-Driven Reinforcement Learning.
> > >
> > > [3] Lipman et al. (2022). Flow Matching for Generative Modeling.
> > >
> > > [4] Lipman et al. (2024). Flow Matching Guide and Code.
> > >
> > > [5] Mnih et al. (2015). Human-level control through deep reinforcement learning.
> > >
> > > [6] Farebrother et al. (2025). Temporal Difference Flows.
> > >
> > > [7] Grill et al. (2020). Bootstrap your own latent: A new approach to self-supervised Learning.

---

> > > > ### Comment · Reviewer_uJ4U · 2025-11-26
> > > > **Thanks the authors for the rebuttal**
> > > >
> > > > I have checked the rebuttal of the authors. I am generally satisfied with the responses. I would like to maintain my positive evaluation.
> > > >
> > > > At the same time, the authors may point out that flow-based policies do not suit every case, such as a very narrow dataset distribution. Knowing when it is better than Gaussian policies could provide guidance for the use of the proposed value flow.

---

### Official Review · Reviewer_V9wD · 2025-10-26

**Soundness:** 3
**Presentation:** 3
**Contribution:** 3
**Rating:** 6
**Confidence:** 3

**Summary:**

This paper proposes Value Flows, a reinforcement learning (RL) framework that models the entire return distribution using flow-matching generative models instead of discretized or quantile-based approximations. The method introduces a distributional flow-matching (DFM) objective that enforces the learned density paths to satisfy the distributional Bellman equation. The authors further derive a flow derivative ODE to estimate the variance of the return distribution, which is used as a confidence weight to reweight the flow-matching loss. Value Flows is evaluated on 37 state-based and 25 image-based benchmark tasks, showing a reported 1.3× average improvement in success rates over prior offline and offline-to-online RL baselines (Table 1, Fig. 3).

**Strengths:**

- Using the flow-matching approach to model the return distribution for each state-action pair is an interesting and intuitively reasonable idea. Flow matching is a more flexible distribution modeling technique that can handle complex distributions, making it more suitable for modeling return distributions in challenging tasks compared to previous distributional RL methods.
- This paper provides a thorough theoretical analysis and derivation of the Value Flows method, and I believe it is a highly effective and reliable approach.
- The experimental section provides substantial evidence supporting the authors’ claims. The return distributions learned by Value Flows are much closer to the ground truth compared to previous distributional RL methods. Moreover, on benchmarks such as OGBench and D4RL, Value Flows demonstrates clear performance advantages over the baselines.
- The writing in this paper is well-organized, and I can easily grasp the information the authors intend to convey.

**Weaknesses:**

- The paper lacks an analysis of the algorithm’s time and GPU memory consumption. It remains unclear whether flow matching requires more computational time and resources compared to previous methods.
- Some parts of the paper’s demonstrations are not entirely rigorous. For example, in Table 1, categorizing C51 under flow policies seems inappropriate to me.

**Questions:**

- C51 is an online distributional RL algorithm. When applying it to the offline setting, was any adaptation made—such as introducing conservative measures—or was it run directly under its original configuration?
- How does the algorithm perform on the classic D4RL MuJoCo benchmarks? Does it also show a clear performance advantage there?
- Is the return distribution visualization in Figure 2 based on samples from the in-distribution dataset, or on out-of-distribution samples collected separately in the environment? If it is in-distribution, could you additionally provide experiments with out-of-distribution samples? I believe that it would be more meaningful if the distribution learned by value flows demonstrates generalization.

---

> ### Author Response · Authors · 2025-11-24
>
> We thank the reviewer for helpful suggestions for improving the paper. The reviewer raised three main questions: (1) the computational time and resource consumption of Value Flows, (2) the performance of Value Flows on the D4RL MuJoCo benchmarks, and (3) the adaptation of C51 to offline settings. We have attempted to address (1) and (2) by conducting additional ablation experiments. For (3), we provide further clarifications below. **Together with the discussion about other questions, does this address the reviewer’s concerns?**
>
> > The paper lacks an analysis the algorithm’s time and GPU memory consumption.
>
> As suggested by the reviewer, we analyzed our experiment logs to further investigate the computational time and GPU memory consumption of different methods in Appendix D.3. Specifically, we analyze experiment logs on one state-based OGBench task ($\texttt{puzzle-4x4-play-singletask-task4-v0}$) and one image-based OGBench task ($\texttt{visual-cube-double-play-singletask-task1-v0}$), comparing Value Flows to selection baselines in Table 3: IQL, ReBRAC, FBRAC, IFQL, FQL, and IQN. We report wall-time and maximum GPU memory usage aggregated across 8 seeds (4 seeds for the image-based task) for different methods. Results in Fig. 10 suggest that training Value Flows requires a computational time similar to training FQL, and the mean wall time of Value Flows is $50\%$ of the mean wall time of IQN. For the maximum GPU usage, Value Flows used substantially less GPU memory than FBRAC and IQN because they directly use the Euler method in the actor loss (FBRAC) or use additional neural networks to encode quantiles (IQN). Taken together, training Value Flows requires intermediate wall time and GPU memory compared with prior methods.
>
> > How does the algorithm perform on the classic D4RL MuJoCo benchmarks?
>
> To further compare the performance between Value Flows and prior offline RL methods, we conduct additional experiments on the set of *6* D4RL antmaze tasks: $\texttt{antmaze-umaze-v2}$, $\texttt{antmaze-umaze-diverse-v2}$, $\texttt{antmaze-medium-play-v2}$, $\texttt{antmaze-medium-diverse-v2}$, $\texttt{antmaze-large-play-v2}$, and $\texttt{antmaze-large-diverse-v2}$. Since these tasks all use state-based observations, we compare Value Flows against selective baselines in Table 3, reporting means and standard deviations over 8 random seeds as in Sec. 5. We bold values within $95\%$ of the best performance for each task.
>
> | Task | BC | IQL | ReBRAC | FBRAC | IFQL | FQL | IQN | Value Flows |
> |:---------------------------:|:-----------------:|:-----------------:|:-----------------:|:-----------------:|:-----------------:|:-----------------:|:-----------------:|:-----------------:|
> |   $\texttt{antmaze-umaze-v2}$    | $55$ | $77$ | $\mathbf{98}$ | $\mathbf{94} \pm 3$ | $92 \pm 6$ | $\mathbf{96} \pm 2$ | $89 \pm 3$ | $\mathbf{94} \pm 3$ |
> |   $\texttt{antmaze-umaze-diverse-v2}$    | $47$ | $54$ | $84$ | $82 \pm 9$ | $62 \pm 12$ | $\mathbf{89} \pm 5$ | $71 \pm 5$ | $80 \pm 4$ |
> |   $\texttt{antmaze-medium-play-v2}$    | $0$ | $60$ | $\mathbf{90}$ | $77 \pm 7$ | $56 \pm 15$ | $78 \pm 7$ | $65 \pm 10$ | $83 \pm 5$ |
> |   $\texttt{antmaze-medium-diverse-v2}$     | $1$ | $74$ | $\mathbf{84}$ | $77 \pm 6$ | $60 \pm 25$ | $71 \pm 13$ | $72 \pm 5$ | $\mathbf{80} \pm 3$ |
> |   $\texttt{antmaze-large-play-v2}$   | $0$ | $42$ | $52$ | $32 \pm 21$ | $55 \pm 9$ | $\mathbf{84} \pm 7$ | $70 \pm 9$ | $\mathbf{86} \pm 6$ |
> |   $\texttt{antmaze-large-diverse-v2}$  | $0$ | $30$ | $64$ | $20 \pm 17$ | $64 \pm 8$ | $\mathbf{83} \pm 4$ | $60 \pm 5$ | $\mathbf{80} \pm 5$ |
>
> On the most challenging tasks $\texttt{antmaze-large-play}$ and $\texttt{antmaze-diverse-play}$, Value Flows achieves a mean normalized return of $83$, where most baselines struggle to reach $70$. While not always outperforming all baselines, the results in this table suggest that Value Flows achieved normalized returns similar to the best-performing baselines on 4 of 6 tasks.

---

> > ### Author Response · Authors · 2025-11-24
> >
> > > C51 is an online distributional RL algorithm. When applying it to the offline setting, was any adaptation made—such as introducing conservative measures—or was it run directly under its original configuration?
> >
> > Yes, we adapted the C51 (and IQN) to the offline RL setting by learning a flow BC policy and using rejection sampling to select actions that maximize the Q estimations from C51 and IQN (see Appendix C.2). This is exactly the reason why we characterized C51 and IQN under the “Flow Policies” category in Table 3. We have revised Appendix E.2 (see the purple text) to include these adaptations.
> >
> > There are two reasons for this adaptation. First, we note that the original C51 [1] and IQN [2] algorithms aim to solve the Atari game with a **discrete** action space, where the greedy action selection is simply taking actions that maximize the Q. However, since our evaluation benchmarks have continuous action spaces, we choose to learn an explicit flow BC policy and use rejection sampling to select actions, resembling the greedy action selection in a discrete action space. Second, using flow-based rejection sampling for C51 and IQN matches the policy extraction method of Value Flows and prevents sampling out-of-distribution actions [3].
> >
> > > Is the return distribution visualization in Figure 2 based on samples from the in-distribution dataset, or on out-of-distribution samples collected separately in the environment?
> >
> > The return distribution visualization in Fig. 2 already uses (temporal) out-of-distribution trajectories. We have revised the text (purple) in Sec. 5.1 and Appendix E.1 to include the following clarifications.
> >
> > Specifically, we use a learned FQL agent to collect 5000 optimal and suboptimal trajectories as our evaluation dataset to visualize the return distribution for Value Flows, C51, and CODAC. These trajectories are out-of-distribution because (1) they are collected by a learned agent instead of the behavioral policy that collected the training dataset, and (2) they yield some combinatorial generalization behaviors. We will clarify the second point below.
> >
> > As mentioned in the prior work [4], the datasets for the set of $\texttt{scene}$ manipulation tasks were collected by scripted non-Markovian policies that
> > randomly interact with different objects in the environment. However, the successful completion of each task requires the agent to sequentially arrange those objects in a specific order (sequential reasoning), which is unseen in the dataset. Hence, to solve the manipulation task in Fig. 2, agents must be able to generalize across trajectories in the dataset (combinatorial generalization).
> >
> > [1] Bellemare et al. (2017). A Distributional Perspective on Reinforcement Learning.
> >
> > [2] Dabney et al. (2018). Implicit Quantile Networks for Distributional Reinforcement Learning.
> >
> > [3] Chen et al. (2022). Offline Reinforcement Learning via High-Fidelity Generative Behavior Modeling.
> >
> > [4] Park et al. (2024). OGBench: Benchmarking Offline Goal-Conditioned RL.

---

> > > ### Comment · Reviewer_V9wD · 2025-11-26
> > >
> > > I thank the authors for their detailed response. The additional experiments and explanations have successfully addressed the concerns and questions raised in my review. I have no further questions, and I will maintain my positive rating.

---

### Official Review · Reviewer_pDXE · 2025-10-27

**Soundness:** 2
**Presentation:** 2
**Contribution:** 2
**Rating:** 2
**Confidence:** 3

**Summary:**

The paper proposes a method to estimate the return distribution using flow matching. To estimate this distribution, the authors leverage a loss based on the distributional Bellman equation and demonstrate how to integrate it into the flow matching loss. This estimate can then be used to train policies either offline or in an offline-to-online setting. The method is validated on state- and image-based tasks from OGBench and D4RL.

**Strengths:**

- As far as I know, the approach is novel.
- I think the topic is interesting and relevant to the community.

However, I believe the weaknesses outweigh the positive aspects by a large margin.

**Weaknesses:**

- The contributions are not clear. The paper presents a loss in the main text and defers the details to the appendix. However, I believe the details are more important than what is stated in the main paper.

**Questions:**

My main concern is about the DCFM loss. The loss seems ill-posed; in particular, $v=0$ is a solution of Equation (5). This loss is not the one used in the experiments, and the reader needs to refer to Appendix C.2 for a clearer understanding. Indeed, in Appendix C.2, the authors mention that the DCFM loss produced divergent vector fields. To address this issue, the authors proposed adding a regularization term. However, I believe the problem is intrinsic to the loss introduced in the main paper. The "regularization" wBCFM seems more sound than the main loss itself.
I think the paper requires at least a major rewrite to properly highlight this "regularization" in the main paper. Similarly, the policy learning aspect should be explained in more detail in the main paper.

Questions:
- Could you comment on the fact that $v=0$ is a solution of Equation (5)? Why should this loss work despite that?
- You tested different regularization coefficients $\lambda$. Could you also train using only the wBCFM? Does it significantly change the performance?

Looking forward to the rebuttal.

---

> ### Author Response · Authors · 2025-11-24
>
> We thank the reviewer for the response and for the suggestions for improving the paper. We have incorporated the reviewer's feedback through revisions to several areas of the paper (purple text in Sec. 4.2 and Appendix C.1) and additional experiments in Appendix D.6, which (1) clarify why Value Flows does not collapse to a trivial solution and (2) the relative importance of the two loss terms. **Together with the discussion below, does this fully address the reviewer’s concerns about the paper?**
>
> > Could you comment on the fact that $v = 0$ is a solution of Equation (5)? Why should this loss work despite that?
>
> We agree that a vector field that simply minimizes the $\mathcal{L}_{\text{DCFM}}$ loss in Eq. 5 might collapse to a constant (e.g., $v = 0$), and have revised Sec. 4.2 and Appendix C.1 (see purple text) to explain why this failure mode does not arise in practice. The target vector field is the main reason for Value Flows not collapsing to deneragated solutions.
>
> This degeneration problem has already existed in prior self-supervised representation learning methods, e.g., BYOL [1]. Specifically, BYOL also learns representations to predict output from the model itself, which might converge to a model ignoring the inputs and always outputting a constant. Instead of using the model from the previous iteration, they use an exponential moving average of the historical models (target model) to predict target outputs. In the experiments of Grill et al. (2020), this target network empirically helps prevent model collapse, and this observation motivates us to use a target vector field in Value Flows (Appendix C.1). Tian et al. (2021) [2] provide a theoretical characterization of how BYOL avoids this failure mode.
>
> > You tested different regularization coefficients $\lambda$. Could you also train using only the wBCFM? Does it significantly change the performance?
>
> As suggested by the reviewer, we ran additional ablations (Appendix D.6) studying the effect of training a Value Flows variant only using the $\mathcal{L}_{\text{wBCFM}}$ loss. We evaluate the success rates on two OGBench tasks $\texttt{cube-triple-play-singletask-task1-v0}$ and $\texttt{scene-play-singletask-task2-v0}$, computing means and standard deviations over $8$ seeds. Results in Fig. 13 suggest that training Value Flows only using the BCFM loss yields trivial (near-zero) success rates on both tasks. In contrast, minimizing the DCFM loss provides the most learning signal for Value Flows. Thus, the BCFM loss mainly serves as a regularization.
>
> [1] Grill et al. (2020). Bootstrap your own latent: A new approach to self-supervised Learning.
>
> [2] Tian et al. (2021). Understanding self-supervised Learning Dynamics without Contrastive Pairs.

---

> > ### Author Response · Authors · 2025-11-27
> >
> > Dear Reviewer,
> >
> > We have worked hard to incorporate the review feedback by running new experiments and revising the paper. Do the revisions and discussions above address your concerns? We would greatly appreciate your engagement.
> >
> > Thanks!
> >
> > The Authors

---

### Official Review · Reviewer_n5xq · 2025-11-01

**Soundness:** 3
**Presentation:** 3
**Contribution:** 3
**Rating:** 6
**Confidence:** 3

**Summary:**

The paper addresses the inability of prior works in distributional RL to model return distributions due to the approximations made by discretizing the distribution. The paper leverages the recent success of flow models to model the full return distribution and better characterize the uncertainty of the return distribution for better flow matching. The authors ensure that at every step, the return distribution follows the distributional Bellman equation. Doing so, they only need to learn the vector field $v$ which they do using a variant of the distributional flow matching loss (which has the same gradient). They perform experiments on a number of state-based and image based domains in both offline RL and offline-to-online RL.

**Strengths:**

(1) The authors build on the recent success of flow models to model full return distributions for distributional RL which can be a major move forward from the discrete assumptions in prior works.

(2) The authors derive a way in which they only need to model the vector field and the flow would follow from it. Seems a clean and nice way to avoid complexities still staying within the theoretical bounds.

(3) The authors propose a flow matching loss that avoids transition dynamics and other complexities. The proposed loss has the same gradient as that of the true loss.

**Weaknesses:**

(1) While the authors mention that distributional RL has advantages in exploration and safety, the domain chosen by them does not seem to be using these properties of distributional RL. To study the effects of modeling the full distribution and uncertainty, a better domain can be chosen.

(2) The preliminaries for flow matching should be better. Currently there is no explanation on how $\phi$ is computed (which is relevant in the later sections of the paper). Similarly, since Lemma 2 is crucial for understanding of the method, it should be in the main paper and not in the appendix.

**Questions:**

(1) Why do you think that aleatoric uncertainty would be better than epistemic?

---

> ### Author Response · Authors · 2025-11-24
>
> We thank the reviewer for the helpful suggestions for improving the paper. It seems like the reviewer's main suggestion is to evaluate our method on a domain crafted to study properties of distributional RL; we have done this by running new experiments using a canonical discrete MDP adapted from Keramati et al. (2019) [3], which requires the policy to do exploration and risk-sensitive control. We have also made revisions to the Preliminaries section to incorporate the feedback and make sure that all notation is defined. **Together with the discussion below about other questions, does this address the reviewer’s concerns?**
>
>
> > To study the effects of modeling the full distribution and uncertainty, a better domain can be chosen.
>
> Thanks for the suggestion. We agree that using a domain that requires exploration and risk-sensitive control to demonstrate the capability of Value Flows. We note that prior distributional RL methods [1, 2] usually evaluate their methods in the standard RL benchmarks, e.g., Atari games. Following these prior methods, we choose to evaluate on the standard offline and offline-to-online RL benchmarks to show the benefits of using Value Flows to estimate Q values in the actor-critic framework. Nevertheless, we include additional experiments on the discrete Machine Replacement MDP from Fig. 2 of Keramati et al. (2019) [3]. We have revised Sec. 5.3 and Appendix D.4 to include these new results.
>
> Following Keramati et al. (2019), we choose a variant of the Machine Replacement MDP (with only $11$ states) to conduct our experiments (Fig.11 (Left)), measuring the $\text{CVaR}\_{0.1}$ performance instead of the expected return. This MDP is especially challenging because of the chain structure, and it also requires risk-sensitive control due to the high variance in the reward distributions at the end of the chain. Therefore, it serves as a good evaluation task for distributional RL algorithms.
>
> Since the machine replacement MDP has a discrete action space, we remove the policy extraction procedure in Value Flows and use the learned vector fields to choose greedy actions directly, resembling the original C51 and IQN implementations. We use a random policy and uniformly initialize the start state to collect a dataset with $100\text{K}$ transitions for offline training. We compare Value Flows against C51 and IQN and use $\text{CVaR}\_{0.1}$ instead of the Q-value ($\text{CVaR}_{0.5}$) to optimize all methods. We compute the means and standard deviations of $\text{CVaR}\_{0.1}$ over 4 random seeds. Results in Fig.11 (Right) show that Value Flows converges to the optimal $\text{CVaR}\_{0.1}$ with a number of gradient steps similar to IQN ($600$), while C51 failed to converge within $5000$ gradient steps. We also observe that Q-learning immediately converges to the optimal expected return and is not able to learn a risk-sensitive policy.
>
> > Why do you think that aleatoric uncertainty would be better than epistemic?
>
> In this paper, we do *not* claim that aleatoric uncertainty is better than epistemic uncertainty or vice versa. The benefits of using aleatoric uncertainty or epistemic uncertainty depend on the applications. For example, if practitioners aim to learn a critic with safety constraints [3, 4], it might be better to consider the aleatoric uncertainty in the environment. In contrast, if the goal of practitioners is to learn an ensemble of critics for stabilizing boostrapped error [5] or covering the entire state space [6], modeling the epistemic uncertainty will be a better choice. Since the distributional RL framework focuses on the uncertainty in the environment, we propose an approach to estimate the aleatoric uncertainty (Sec. 4.3).

---

> ### Author Response · Authors · 2025-11-24
>
> > Currently there is no explanation on how $\phi$ is computed (which is relevant in the later sections of the paper). Similarly, since Lemma 2 is crucial for understanding of the method, it should be in the main paper and not in the appendix.
>
> We have made revisions to the Preliminaries (see the purple text in Sec. 3) to clarify these details. The $\phi$ is the time-dependent diffeomorphic flow that transforms the Gaussian noise into the target data. This flow function is the core and a standard component of the flow matching generative models [7, 8]. As mentioned in Appendix A.3, the flow $\phi$ and the vector field $v$ follows the ODE,
> \begin{align*}
>  \frac{d}{d t} \phi(\epsilon \mid t) = v(\phi(\epsilon \mid t) \mid t),\quad \phi(\epsilon \mid 0) = \epsilon,\quad \phi(\epsilon \mid 1) = \hat{x}.
> \end{align*}
> In practice, we fit the vector field using flow-matching objectives and use the Euler method to iteratively compute the flow at any time step.
>
> Due to the page limit, our initial submission deferred the introduction of the flow matching into Appendix A.3 and Lemma 2 (regarding density of return distributions) into Appendix A.2. We are happy to bring them back in the final version of the paper. Please let us know if there are specific paragraphs that are still difficult to understand. We would be happy to incorporate your feedback to further improve the paper.
>
> [1] Bellemare et al. (2017). A Distributional Perspective on Reinforcement Learning.
>
> [2] Dabney et al. (2018). Implicit Quantile Networks for Distributional Reinforcement Learning.
>
> [3] Keramati et al. (2019). Being Optimistic to Be Conservative: Quickly Learning a CVaR Policy.
>
> [4] Delage and Mannor (2010). Percentile Optimization for Markov Decision Processes with Parameter Uncertainty.
>
> [5] Lee et al. (2020). SUNRISE: A Simple Unified Framework for Ensemble Learning in Deep Reinforcement Learning.
>
> [6] Pathak et al. (2017). Curiosity-driven Exploration by Self-supervised Prediction.
>
> [7] Lipman et al. (2022). Flow Matching for Generative Modeling.
>
> [8] Lipman et al. (2024). Flow Matching Guide and Code.

---

> > ### Author Response · Authors · 2025-11-27
> >
> > Dear Reviewer,
> >
> > We have worked hard to incorporate the review feedback by running new experiments and revising the paper. Do the revisions and discussions above address your concerns? We would greatly appreciate your engagement.
> >
> > Thanks!
> >
> > The Authors

---

### Author Response · Authors · 2025-12-02
**Discussion summary**

Dear AC,

We sincerely appreciate your time and effort in evaluating this work during this exceptional situation. In case it's helpful, we would like to provide a summary of the discussions so far.

We have worked hard to incorporate the reviewers’ feedback with additional experiments and revisions. Before the reviews were rolled back, the most recent ratings (as of Nov 27) were **6, 2, 6, 6** from reviewers n5xq, pDXE, V9wD, and uJ4U, respectively. We received replies from reviewers V9wD and uJ4U confirming that we addressed their concerns. They explicitly stated to maintain their positive ratings (6). However, we had **not** received follow-ups from reviewers n5xq and pDXE, making it unclear whether our clarifications reached them. Below, we summarize their concerns and our responses.

Reviewer n5xq gave a positive initial rating (6) and raised two suggestions: (1) evaluate Value Flows on a domain crafted to study properties of distributional RL, and (2) clarify important concepts and lemmas in the preliminary section. We have attempted to address (1) by running new experiments (Appendix Fig. 11) using a canonical discrete MDP adapted from Keramati et al. (2019) [1]. For (2), we have revised the Preliminary (see the purple text in Sec. 3) to incorporate the feedback and make sure that all notation is defined. We believe [our](https://openreview.net/forum?id=2VyNYUVF2k&noteId=N9A3CL9mc4) [responses](https://openreview.net/forum?id=2VyNYUVF2k&noteId=t1HHwXtBzp) have fully addressed these two suggestions.

Among all the reviewers, reviewer pDXE is the only one who gave a negative rating of 2, as of Nov 27. Reviewer pDXE raised some important questions about (1) whether our method collapses to a trivial solution, and (2) the relative importance of the two loss terms $\mathcal{L}_ \text{wDCFM}$ and $\mathcal{L}_ \text{wBCFM}$. Indeed, as we've revised the paper to clarify, a trivial solution is admissible by our objective. However, this trivial solution is not found in practice. We have attempted to address (1) by mentioning a prior method, BYOL [2], that incurs the same issue. Our method adopts a target network during training, similar to BYOL, which avoids the failure mode of collapsing. We have revised several areas of the paper (purple text in Sec. 4.2 and Appendix C.1) to incorporate these discussions. For (2), we have included additional experiments in Appendix D.6 showing that the DCFM loss provides the most learning signal for Value Flows, while the BCFM loss mainly serves as a regularization. We believe that [our responses](https://openreview.net/forum?id=2VyNYUVF2k&noteId=PVMyhJnZTR) have fully addressed these two concerns.

We have included new experimental results in the revised paper and incorporated specific writing suggestions mentioned by the reviewers. We hope that this summary helps assess our submission. Please feel free to let us know if there are any other concerns or questions. Thank you again for your time and effort in reviewing our work under these unusual circumstances.

The Authors

[1] Keramati et al. (2019). Being Optimistic to Be Conservative: Quickly Learning a CVaR Policy.

[2] Grill et al. (2020). Bootstrap your own latent: A new approach to self-supervised Learning.

---

### Meta-Review · Area_Chair_RRWQ · 2026-01-04

**Summary:**

The paper proposes "Value Flows," applying flow-matching to estimate return distributions in RL. Three reviewers (n5xq, V9wD, uJ4U) supported the paper, citing its novelty and SOTA performance across extensive benchmarks. One reviewer (pDXE) raised concerns about potential trivial solutions and the placement of key formulas in the appendix. The authors added further clarification to the "trivial solution" concern during rebuttal by explaining the role of target networks (similar to previous works) and providing ablation studies. The performance gains are promising and the methodology is promising. The structural issues raised are not addressed during the rebuttal but can be easily fixed in the final version.

The AC therefore recommends acceptance. However, please address the following points and reviewers' suggestions in the final version:
1. The practical loss function and the Bootstrapped Conditional Flow Matching (BCFM) regularization term (currently in Appendix C.1) should be moved to the main text. The method description is incomplete without explaining this regularization.
2. Summarize the findings regarding the method's performance on narrow data distributions (e.g., D4RL Adroit) in the main text's discussion/limitation section.
3. Incorporate the intuition regarding the target network preventing collapse into the methodology section to further clarify the theoretical point.

**Reviewer Concerns:**

**Reviewer n5xq**
(Addressed) Concern 1: Choice of experimental domain
- The reviewer pointed out that while the paper claims advantages in exploration and safety (typical of distributional RL), the chosen domains (standard offline/offline-to-online benchmarks) did not adequately showcase these specific properties. The reviewer suggested a “better” domain to study the effects of modeling the full distribution and uncertainty.
- The authors ran new experiments on a canonical discrete MDP (Machine Replacement MDP from Keramati et al., 2019) designed to require exploration and risk-sensitive control. They compared the proposed method (Value Flows) against C51 and IQN using CVaR optimization. The results showed Value Flows converged to the optimal CVaR while baselines struggled.

(Addressed) Concern 2: Clarity of preliminaries (Flow Matching & Lemma 2)
- The reviewer pointed out that the explanation of flow matching was insufficient. Also, Lemma 2, which is crucial for understanding, was buried in the appendix.
- The authors revised the paper to define $\phi$ and the ODE explicitly, and committed to the structural change requested.

(Addressed) Concern 3: Aleatoric vs. epistemic uncertainty
- The reviewer asked why the authors believe aleatoric uncertainty would be better than epistemic.
- The authors Clarified that they do not claim one is strictly better than the other; it depends on the application (e.g., safety constraints vs. exploration/ensemble stability). They justified their focus on aleatoric uncertainty because distributional RL frameworks typically focus on environmental uncertainty.
---
**Reviewer pDXE**
(Partly addressed) Concern 1: Ill-posed loss function
- The reviewer raised a major concern that the main loss function (DCFM) is ill-posed because a vector field of zero ($v=0$) is a valid solution to the equation. They questioned why the method works despite this problem.
- The authors acknowledged that minimizing the loss could theoretically collapse to a constant. However, they argued that in practice, they use a target network (similar to BYOL in self-supervised learning) which prevents this collapse. They revised the text to explicitly explain this mechanism.

(Partly addressed) Concern 2: The role of regularization
- The reviewer believed the "regularization" term (wBCFM) found in the appendix was actually the sound part of the method, while there are some problems in the main loss (DCFM). The reviewer suspected the method only worked because of the regularization and requested a major rewrite to highlight wBCFM as the primary component, and asked if the model could train using only wBCFM.
- The authors performed the requested ablation study (training with only wBCFM), which showed that success rates dropped to near zero when using only the regularization. This empirically demonstrated that the DCFM loss is the primary learning signal.

(Not properly addressed but can be easily fixed in the final version) Concern 3: Paper structure and clarity
- The reviewer felt the paper hid important details (like the regularization and policy learning specifics) in the appendix and requested a major rewrite.
- The authors mentioned that they revised the main text (Section 4.2) to clarify the loss terms and the non-collapse mechanism. However, the BCFM loss part is still not mentioned in the main text. Since this is also critical to stabilize learning, it is critical to also mention this in the main text.
---
**Reviewer V9wD**
(Addressed) Concern 1: Computational efficiency
- The reviewer noted a lack of analysis regarding time and GPU memory consumption, expressing concern that flow matching might be computationally expensive compared to prior methods.
- The authors provided a new analysis comparing wall-clock time and GPU usage, where they demonstrated that Value Flows has training times comparable to FQL and is faster than IQN, while using intermediate GPU memory.

(Addressed) Concern 2: Performance on standard benchmarks
- The reviewer asked how the algorithm performs on classic D4RL MuJoCo benchmarks to see if the performance advantage holds there as well.
- The authors added experiments on 6 D4RL Antmaze tasks. The results showed that Value Flows is competitive, achieving the best or near-best performance on the most challenging tasks (large-play and large-diverse).

(Addressed) Concern: Baseline implementation & categorization

- The reviewer questioned the categorization of C51 (a discrete distributional algorithm) under Flow Policies in the paper's tables and asked how it was adapted for the offline setting.
 - The authors clarified that they adapted C51 and IQN by learning a flow-based behavioral cloning policy and using rejection sampling to select actions. This specific adaptation justifies the categorization. They updated the appendix to reflect these details.

(Addressed) Visualization & Generalization
- The reviewer asked if the return distribution visualizations (Figure 2) used in-distribution or out-of-distribution (OOD) samples, suggesting that OOD samples would better demonstrate generalization.
- The authors clarified that the figure already used OOD trajectories collected by a separate learned agent, not the training data. They revised the text to make this explicit.
---
**Reviewer uJ4U**
(Addressed) Concern 1: Clarity and presentation
- The reviewer found the method section difficult to follow due to dense notation and implicit logic. They also noted that the paper was not self-contained because the complete loss function was relegated to the Appendix.
- The authors added a notation table (Table 2) and improved signposting in Section 4 to clarify the logical flow. They also committed to moving the complete loss function back into the main text for the camera-ready version.

(Addressed) Concern 2: Performance on D4RL adroit tasks
- The reviewer asked why Value Flows performed worse than baselines (ReBRAC, IQL) specifically on D4RL Adroit tasks.
- The authors hypothesized that flow-based policies struggle with the "narrow dataset distribution" characteristic of Adroit, and ran an ablation replacing the flow-based policy with a Gaussian policy, which improved performance to match state-of-the-art baselines.

(Addressed) Concern 3: Computational efficiency & hyperparameters
- The reviewer requested a comparison of compute efficiency and asked how reducing flow steps (to 1 or 2) affects performance.
- The authors provided logs showing Value Flows has training times similar to FQL and is much faster than IQN, with intermediate memory usage; and also provided ablation results showing that while 1-2 steps work for simple tasks, complex tasks suffer a significant performance drop, justifying their choice of 10 steps.

Concern 4: Technical concepts (vector fields)
- The reviewer asked for better explanations of the vector field's role and the necessity of the "target return vector field."
- The authors provided the mathematical intuition (ODE/velocity) and explained that the target vector field is a standard technique (EMA) used to stabilize learning and prevent model collapse, similar to methods like BYOL.

**Reviewer Scores:**

- Reviewer n5xq: maintain positive
- Reviewer pDXE: would likely increase to a 4 (from 2). The reviewer gave a strong reject (2) largely based on the hypothesis that the method contained an ill-posed loss function and that some of the parts were misrepresenting which loss function was actually doing the work. The authors provided further clarification about the trivial solution part, where the authors used a target network as in previous work to prevent the collapse (explained the $v=0$ issue using standard literature (BYOL)). The authors also ran the requested ablation study, which showed that the regularization alone cannot solve the task. While the technical concerns were answered, the reviewer might still feel the paper's presentation is too "fair" as the concerns were not fully addressed (but can be further fixed in the final version).
- Reviewer V9wD: maintain positive (as stated in the reviewer’s final comment)
- Reviewer uJ4U: maintain positive (as stated in the reviewer’s final comment)

---

### Decision · Program_Chairs · 2026-01-26

Accept (Poster)